

**Spatio-temporal variability and light absorption property of carbonaceous aerosol in a typical glacierization region of the Tibetan Plateau**

Hewen Niu[a], Shichang Kang[a,c,*], Hailong Wang[b,*], Rudong Zhang[d,b], Xixi Lu[e], Yun Qian[b], Shijin

Wang[a], Rukumesh Paudyal[a], Xiaofei Shi[f] and Xingguo Yan[g]

[a] State Key Laboratory of Cryospheric Science, Northwest Institute of Eco-Environment and

Resources, Chinese Academy of Sciences, Lanzhou 730000, China.

[b] Atmospheric Sciences and Global Change Division, Pacific Northwest National Laboratory

(PNNL), Richland, WA 99352, USA

[c] CAS Center for Excellence in Tibetan Plateau Earth Sciences, Beijing 100101, China.

[d] Institute for Climate and Global Change Research, School of Atmospheric Sciences, Nanjing

University, Nanjing, China; Collaborative Innovation Center of Climate Change, Jiangsu Province,

China.

[e] Department of Geography, National University of Singapore, 1 Arts Link, Singapore 117570,

Singapore

[f] College of Earth Environmental Sciences, Lanzhou University, Lanzhou 730000, China.

[g] College of Geography and Environmental Science, Northwest Normal University, Lanzhou

730030, China

* Corresponding authors: shichang.kang@lzb.ac.cn; Hailong.Wang@pnnl.gov





**Abstract:**

   The high altitude glacierized regions of the Tibetan Plateau (TP) are influenced by carbonaceous aerosols from local sources and long range transport from the adjoining areas. Deposition and accumulation of light-absorbing carbonaceous matters on
glacier surfaces can alter the energy balance of glaciers. In this study, two years (December 2014 to December 2016) of continuous observations of carbonaceous aerosols performed in glacierized region of Mt. Yulong (4510 m asl) and Ganhaizi (GHZ) basin (3054 m asl) are analyzed. The mass absorption efficiency (MAE) of black carbon (BC) was determined for the first time in Mt. Yulong using a
thermal-optical carbon analyzer. The average BC and organic carbon (OC) concentrations were $1.51 \pm 0.93$ and $2.57 \pm 1.32$ μg m$^{-3}$, respectively. The average SOC (secondary OC) concentration, quantified using BC-tracer method, was $1.67 \pm 1.15$ μg m$^{-3}$. Monthly mean BC concentrations from monsoon to post-monsoon season were higher than OC in the high altitudes (approximately 5000 m asl) of Mt. Yulong. The
concentrations of carbonaceous matter have distinct spatial and inter-annual variations in this glacierization area. High carbonaceous matter associated with OC (including both SOC and POC) in GHZ basin was mainly contributed from tour bus emissions. The annual mean OC/BC ratio was $2.45 \pm 1.96$ in Mt. Yulong. Strong photochemical reactions and local tourism activities in monsoon season were the main factors
inducing high OC/BC ratios in the Mt. Yulong region. The mean MAE of BC, measured at 632 nm with a thermal-optical protocol under the filter-based method, was $6.82 \pm 0.73$ m$^2$ g$^{-1}$, comparable with the results from other studies. Strong seasonal and spatial variations of BC MAE were largely related to the OC and SOC abundance. Source attribution analysis using a global aerosol-climate model, equipped with a BC
source tagging technique, suggests that East Asia emissions, including local sources, have the dominant contribution (over 50%) to annual mean near-surface BC at the two sites. There is also a strong seasonal variation in the regional source apportionment. South Asia has the largest contribution during the pre-monsoon season, while East Asia dominates the monsoon season and post-monsoon season. Results in this study
have great implications for accurately evaluating the influences of carbonaceous





matter on glacial melting and water resource supply in glacierization areas.

*Keywords*: Black carbon; Carbonaceous aerosol; Light absorption; Mt. Yulong

## 1. Introduction

Carbonaceous aerosols play an important role in Earth's climate system and energy budget (Bond et al., 2013; Schuckmann et al., 2016). It has sophisticated/complex effects on the health of human and living organisms (Jerret et al., 2005), visibility, atmospheric radiative balance, and the surface albedo of snow and ice. Carbonaceous aerosol is an aggregate of thousands of species with various thermal, physicochemical

and optical properties (Andreae and Gelencsér, 2006; Cheng et al., 2011a). In the atmosphere, carbonaceous aerosols affect the radiative balance by absorbing and scattering solar radiation and affecting the properties of clouds (IPCC, 2013; Lohmann and Feichter, 2005; Carslaw et al., 2010). In the cryosphere, deposition of carbonaceous matter on snow and glaciers reduces the surface spectral albedo (snow

darkening) (Flanner et al., 2009; Doherty et al., 2013; Qian et al., 2015; Niu et al., 2017a) and accelerates snow/glacial melting (Hansen and Nazarenko, 2004; Xu et al., 2009a).

    Carbonaceous matter in smoke/emissions from biomass burning and fossil fuel combustion has been identified as the typical atmospheric pollutant since historical

period (Brimblecombe, 1987). Recently, scientific attention has shifted from the role of carbonaceous matter as atmospheric pollutant to its influence as one of driving factors of climate change (Andreae, 1995; Andreae and Gelencsér, 2006; Hansen et al., 2005; Ramanathan et al., 2005). Some model simulations proposed that the radiative forcing of black carbon (BC) is comparable to that of methane (Chung and Seinfeld,

2005; Jacobson, 2004), suggesting that BC may be the second most important warming agent (only after $CO_2$) in terms of direct radiative forcing (Jacobson, 2001). Ding et al. (2016) found that BC particles play a key role in modifying/heating the planetary boundary layer (PBL) and enhancing the haze pollution, called the "dome effect" of BC, and suggested an urgent need for reducing BC emissions to mitigate the

extreme haze pollution in megacities in China. In addition, high concentrations of



absorbing aerosols over eastern China during winter and spring were found to be the major reason for the observed recent warming trend (Yu et al., 2001). BC in snow can increase the surface air-temperature by approximately 1.0 °C over the Tibetan Plateau (TP) and reduce spring snow cover (Qian et al., 2011). Generally, sampled

carbonaceous aerosols can be divided into elemental carbon (EC) and organic carbon (OC) using thermal/optical reflectance (TOR) method (Cao et al., 2010; Chow et al., 1993), and EC is usually used synonymously with BC (Chen et al., 2011; Ming et al., 2013; Xu et al., 2009b). Moreover, in the low-latitude and high-elevation areas, extensive incoming solar radiation and large amount of carbonaceous aerosol

deposited on snowpack and glaciers result in surface albedo reduction and the retreat of glaciers in the TP, and further affect Asian hydrological cycle and monsoon climate (Qian et al., 2011; Qu et al., 2014; Li et al., 2016a). This is closely related to water resources for billions of local habitants in South Asia (Ramanathan et al., 2005, 2007). Therefore, it is rather important and necessary to carry out carbonaceous aerosol study

in glacierization regions.

The mass absorption efficiency (MAE, $m^2\,g^{-1}$) is a typical parameter characterizing the optical (or light absorbing) properties of aerosols. For BC particles it is determined by the mass concentration ($\mu g\,m^{-3}$) and absorption coefficient ($b_{abs}$, $Mm^{-1}$) of BC (Liousse et al., 1993; Chen et al., 2011), where $b_{abs}$ is the cross section of BC

available to absorb light (Bond and Bergstrom 2006; Knox et al., 2009). MAE of BC is usually estimated using quartz filter based methods, which detect the change in the light transmittance through a quartz filter due to the presence of BC particles (Sharma et al., 2002; Knox et al., 2009; Cheng et al., 2011a). Bond and Bergstrom (2006) suggested a mass-normalized MAE of $7.5\pm1.2\ m^2\,g^{-1}$ at 550 nm for uncoated aerosol

particles. However, Ram and Sarin (2009) studied the $b_{abs}$ and MAE of BC in aerosols sampled at three typical sites in India, and they found a distinct spatio-temporal variability in $b_{abs}$ and BC MAE on a regional scale. Discrepancies are sometimes inevitable for the filter based techniques related to aerosol-filter interactions (Cheng et al., 2011a; Ssandradewi et al., 2008). Moreover, the MAE of BC can be largely

influenced by the aerosol mixing state (Bond et al., 2006; Doran et al., 2007; Jacobson,





2001; Schnaiter et al., 2005). It was proposed that non-carbon species (e.g., sulphate, nitrate) can increase the MAE value of BC (Knox et al., 2009) because the coating by other components can focus light into the BC core of the particle (Bergstrom et al., 1982; Cheng et al., 2011a). Enhancement of MAE by coating can be described in terms of absorption amplification that is largely independent of coating thickness (Schnaiter et al., 2005; Knox et al., 2009). Therefore, it is quite necessary to make a further analysis on BC MAE to reduce the uncertainties in evaluating light absorption of carbonaceous aerosols.

Recently, in addition to BC, it has been recognized that certain organic carbon (OC) components in carbonaceous aerosol can also absorb light (Andreae and Gelencsér, 2006; Cheng et al., 2011a). However, light absorption by OC has not yet been taken into consideration in most climate models, which causes certain uncertainties in precisely evaluating climate/radiative forcing of carbonaceous aerosol in the atmosphere and snow/ice. The radiative forcing of carbonaceous aerosol remains one of the great challenges in climate simulation (Jacobson, 2001).

In this study, total suspended particulates (TSP) were simultaneously collected at two remote stations on Yulong Snow Mountain (abbreviated Mt. Yulong), in the southeast fringe of the TP. Spatial and seasonal variations of atmospheric carbonaceous matter are examined, and the corrected BC MAE is calculated to evaluate light-absorbing property of carbonaceous aerosol in the typical glacierization area. Cloud-Aerosol Lidar with Orthogonal Polarization (CALIOP) is used to retrieve total attenuated backscatter and sub-type information of smoke plumes to help on source attribution of carbonaceous aerosols in Mt. Yulong.

**2. Material and methods**

2.1 Study area and TSP sampling

The study area, Mt. Yulong (26°59′-27°17′ N, 100°04′-100°15′ E), is the southernmost glaciated mountain in the Eurasian continent (Fig. 1). Mt. Yulong is exposed to emissions from South Asia and impacted by carbonaceous aerosol during Indian summer monsoon more than other regions in the TP. The Baishui glacier (27°06'16″ N; 100°11'44″ E) on Mt. Yulong is a typical temperate glacier that has





large energy fluxes, particularly at lower snow-covered elevations. The climate of Mt. Yulong is typically affected by the Indian summer monsoon and East Asian summer monsoon (Nie et al., 2017) in the monsoon season (June-September), and characterized by other three distinct seasons: post-monsoon (Oct. - Nov.), winter (Dec. - Feb.), pre-monsoon (Mar. - May) season (Chen et al., 2015; Cong et al., 2015a; Bonasoni et al., 2010; Niu et al., 2013). Annual rainfall in Mt. Yulong occurs considerably (~80%) in the monsoons season (Fig. 2). There is little rainfall in winter when the westerly jet dominates (Liu et al., 2017).

One of the TSP sampling sites is at the elevation of 4510.25 m asl, which is close to the location of the upper station (27°06′16.953″ N, 100°11′59.297″ E) of tourism cableway on Mt. Yulong. Ganhaizi (GHZ) basin, the other TSP sampling site (27°06′08.285″ N, 100°15′25.418″ E), is located on the east side and the foot of Mt. Yulong, at an elevation of 3054.05 m asl. It is separated from the urban area, but is an increasingly popular tourist destination, with a geological museum and a golf course. Moreover, there is a spacious parking lot and a tourist dining-center in GHZ. Beside the emission from tourist vehicles, there are very limited pollution source in the study sites. These two sampling sites are located on the southeast fringe of the TP, away from urban areas, and thus considered as typical remote areas in the Northern Hemisphere (Li et al., 2016a) and ideal observation areas for atmospheric environment in glacierization region.

The TSP samples analyzed in this study were collected from December 2014 to December 2016 at these two sites using a particulate sampling apparatus (TH150-A, Wuhan Tianhong INST Group). The sampling apparatuses were placed 15 m above the ground, away from the surface dust and any specific pollutant sources. Atmospheric air (and suspended particulates) was sampled at a flow rate of 100 L min$^{-1}$ and each sample was collected for 24 h using a stable vacuum pump, the sampling interval for each TSP sample was 6 days. The collected TSP samples were loaded on 90 mm (in diameter) pre-combusted (heating at 550 ℃ for 6 h in an oven) quartz fibre filters (Whatman Corp.).

After sampling, the quartz filters were wrapped with an aluminum foil and were





kept in a refrigerator at 4 ℃ in the Yulong Glacier Station of Chinese Academy of Sciences in Lijiang city, and then were transported to the State Key Laboratory of Cryospheric Science in Lanzhou city for chemical analysis. In addition, precautions were taken during both collection and analysis processes to avoid the possible

contamination of TSP samples.

2.2 Sample analysis

OC and BC on the quartz filters were analyzed using a Desert Research Institute (DRI) Model 2001 thermal-optical reflectance carbon analyzer (Atmoslytic Inc., Calabasas, California) to determine the BC concentration and optical attenuation

(ATN) (Chow et al., 2001; Cheng et al., 2011a; Li et al., 2016b; Niu et al., 2017a). Every filter was analyzed for a portion of carbon in a 0.502 $cm^2$ punch. A temperature peak (550 ℃) was designed to reduce measuring time that BC exposed in the catalyzing atmosphere. The applied heating conditions permitted the separation of BC portions in $O_2$ (2%) and helium (98%) atmosphere and OC portions in a helium

atmosphere (Wang et al., 2015; Niu et al., 2017a).

2.3 Calculation methods of secondary OC (SOC), ATN and MAE

2.3.1 SOC calculation

$$OC_{pri}= (BC \ (OC/BC)_{min}, OC_{tot})_{min} \tag{1}$$

$$OC_{sec}= OC_{tot}-OC_{pri} \tag{2}$$

where $(OC/BC)_{min}$ is the minimum OC/BC ratio in a specific set of data, $OC_{pri}$ indicates the primary OC (or POC) content. SOC (i.e. $OC_{sec}$) is calculated as the difference between total OC content ($OC_{tot}$) and $OC_{pri}$. The $OC_{pri}$ is constrained so that it may not exceed the $OC_{tot}$ (Yu et al., 2004, 2007). The minimum OC/BC (OC-to-BC) ratio was suggested for use in the calculation of SOC concentration in

several previous studies (Castro et al., 1999; Cong et al., 2015a; Ram and Sarin, 2015; Pachauri et al., 2013). Semi-empirical BC-tracer technique was employed in the SOC calculation, and $OC_{sec}$ was set to zero in cases where $(OC/BC)_{pri} > (OC_{tot}/BC)$ (Yu et al., 2007). These cases are indicative of model overestimations of $(OC/EC)_{pri}$. The semi-empirical estimates of $OC_{sec}$ after screening for outliers are comparable to the

empirical estimates, and sometimes are even superior to empirical estimates.



Nonetheless, it should be noted that both semi-empirical $OC_{sec}$ estimates and empirical estimates have limitations, and there is no standard or more perfect method of $OC_{sec}$ estimation (Yu et al., 2007). Thus, BC-tracer method employed here is reliable in determining SOC concentrations.

2.3.2 ATN calculation

The ATN is calculated based on the transmittance signal during the filter analysis, defined as

$$ATN = \ln(\frac{I_0}{I})$$

where $I_0$ and $I$ are the transmittance signal after and before the thermal/optical

analysis (Ram and Sarin, 2009). Lights passing through a particle-loaded and a blank filter were simultaneously measured in the determination of ATN by the (thermal-optical reflectance) carbon analyzer. ATN determined by the carbon analyzer is similar to the Aethalometer (Cheng et al., 2011a). Previous studies have demonstrated that ATN of blank quartz fibre filters averaged $0.00\pm0.01$, suggesting

that the ATN of loaded quartz filter could generally be ascribed to the existence of light-absorbing carbon (Yang et al., 2009; Cheng et al., 2011a).

2.3.3 MAE calculation

The mass absorption efficiency (MAE) is calculated as:

$$MAE = \frac{ATN}{BC_s} \times 10^2 \times \frac{1}{C}$$

where $BC_s$ ($\mu g$ C $cm^{-2}$) is the filter loading amount of BC, which is directly measured from the thermal optical analysis. The filter-based determination of light absorption has many artifacts, though the various scattering effects can be properly corrected by an empirical factor, $C$. A value of $C = 3.6$ was proposed for the internally mixed atmospheric aerosol when employing thermal-optical analysis method in several

studies (Weingartner et al., 2003; Doran et al., 2007; Ram and Sarin, 2009; Cheng et al., 2011a). The same empirical factor was also used in the optical measurement by the Aethalometer (Ram and Sarin, 2009). There are many factors (e.g., measurement methods, mixed states) accounting for the discrepancy of MAE. The corrected



equation of MAE calculation (i.e., corrected for the multiple scattering effects) performed in this study has greatly diminished the uncertainties (around 15%). By converting previously published MAE values (Ram and Sarin, 2009) to the equivalent-MAE, Cheng et al. (2011a) has found that the equivalent-MAE was much

lower in the regions severely affected by biomass burning (e.g., Allahabad, India).

The dependence of light attenuation (ATN) detected at 632 nm on BC loading ($BC_s$) is shown in Fig. 3, to identify the artifact relevant to filter-based measurements. As a result, ATN and $BC_s$ correlate well ($R^2$=0.83) with a slope ($K$) of 0.08 m$^2$ g$^{-1}$ and an intercept ($b$) of 0.35 m$^2$ g$^{-1}$ for our samples at the Mt. Yulong sampling site. Strong

correlation ($R^2$=0.62) between ATN and $BC_s$ also exists for the GHZ sampling site, with $K$=0.08 m$^2$ g$^{-1}$ and $b$=0.45 m$^2$ g$^{-1}$. Strong dependence of ATN on $BC_s$ has been found in the present study, and artifact associated with filter-based method was not identified and thus can be neglected.

2.4 Model experiment

We use a global aerosol-climate model, the Community Atmosphere Model version 5 (CAM5) equipped with a BC source tagging technique (Wang et al., 2013, 2014; Zhang et al., 2015), to help estimate the source attribution of BC measured at the two sites. The 4-mode modal aerosol scheme of CAM5 recently developed by Liu et al. (2016) is used here, in which BC and primary OC particles are emitted into a

primary-carbon mode. They then grow through condensation of gas-phase precursors (e.g., sulfuric and organic gases) and move to the accumulation size mode, where hygroscopic aerosol particles, including carbonaceous aerosols, are subject to wet removal by precipitation.

The CAM5 experiment is conducted for five years (2010-2014) at horizontal grid

spacing of 1.9 ° × 2.5 ° and 30 vertical levels. The sea surface temperatures and sea ice concentrations are prescribed with observations, and winds are constrained with reanalysis from NASA Modern Era Retrospective-Analysis for Research and Applications (MERRA) (Rienecker et al., 2011; Ma et al., 2013). Monthly mean anthropogenic and open fire emissions (Hoesly et al., 2017; van Marle et al., 2017;

Yang et al. 2017), including primary OC, SOC gases and BC, used in the simulation



come from the recently released datasets for the Coupled Model Intercomparison Project Phase 6 (CMIP6), which are only available up to year 2014. Therefore, the model experiment is not designed to simulate the two-year observations of aerosols in Tibetan Plateau, but rather for a recent time period (2010-2014) to estimate the mean source attributions and seasonal variations of near-surface BC concentrations.

## 3. Results and discussion

3.1 Characteristics of the carbonaceous aerosols

Temporal variations of carbonaceous matter measured from Mt. Yulong are shown in Fig. 4. Distinct seasonal differences presented during the sampling time period. The winter season and post-monsoon season had higher concentrations of carbonaceous matter in the TSP during the two years, which is consistent for BC, OC, SOC, and POC. However, the OC/BC ratio showed an opposite seasonal contrast. Monsoon season in 2016 had the lowest carbonaceous matter contents in the two-year time period, whereas the concentrations of OC, BC, and POC in the monsoon and pre-monsoon season in 2015 presented relatively high and low values, respectively. This is somewhat uncharacteristic compared with the general seasonal variations of atmospheric chemistry (Kang et al., 2004, 2007; Niu et al., 2013, 2016). It is quite likely that the below-average rainfall and sporadic dust events during the monsoon in 2015 is responsible for this unusual phenomenon (i.e. relatively high content in monsoon season in 2015).

In comparison to Mt. Yulong, the trends of carbonaceous matter from the GHZ basin presented more distinct seasonal variations (Fig. 5), i.e., the monsoon season regularly had the lowest concentrations of BC, OC, SOC, and POC. The OC/BC ratio was consistently opposite to that, e.g., higher values of OC/BC appeared in monsoon seasons. Lower ratio of OC/BC in the other seasons (winter, pre-monsoon, post-monsoon) was probably attributed to the little material available for external-coating of BC, and the limited photochemical ability of OC during the non-monsoon season (Knox et al., 2009). In addition, seasonal changes in BC and OC sources (e.g., biomass burning vs. fossil fuel combustion) might play an important role for the variations of OC/BC ratios.

Table 1 summarizes statistical results of BC, OC, SOC, and POC concentrations in atmospheric aerosol from Mt. Yulong. The annual mean BC concentration was 1.53 ±



1.49 µg m$^{-3}$, with the sampled values ranging from 0.02 to 6.83 µg m$^{-3}$ during the study period (December 2014-Decmber 2016). The post-monsoon season had the highest BC content of 3.51 ±1.2 µg m$^{-3}$, ranging from 1.22 to 5.8 µg m$^{-3}$. The annual average OC and SOC concentrations were 1.65 ±1.14 µg m$^{-3}$ and 1.15 ±0.96 µg m$^{-3}$,

respectively, with the lowest concentrations in the monsoon season (OC: 1.34 ±0.9 µg m$^{-3}$ and SOC: 0.96 ± 0.8 µg m$^{-3}$) and the highest concentrations in the post-monsoon season (OC: 2.57 ±1.3 µg m$^{-3}$ and SOC: 1.42 ±1.2 µg m$^{-3}$). Similar seasonal differences were also found in other areas such as Lumbini, Nepal (Wan et al., 2017), NCO-P (Bonasoni et al., 2010), Kanpur, India (Ram et al., 2012), and

Delhi, India (Mandal et al., 2014). Moreover, monthly averaged carbonaceous aerosol (BC, OC) concentrations were analyzed for Mt. Yulong (Fig. 6). It shows that from the monsoon season to the post-monsoon season (particularly from Sep. to Dec.), BC concentrations were higher than that of OC. A large amount of biomass burning emissions in the high atmosphere (around 5000 m asl) in Mt. Yulong were probably

long range transported from source regions.

Statistical results of BC, OC, SOC, and POC concentrations in aerosol from the GHZ site are shown in Table 2. The annual average BC concentration, 1.5 ±0.9 µg m$^{-3}$, is slightly lower than that from Mt. Yulong, BC deposition in high elevation areas of Mt. Yulong was primarily due to the long-range transport of biomass burning

from South Asia by crossing the Himalayas (Bonasoni et al., 2010; Lüthi et al., 2014; Cong et al., 2015a, b; Li et al., 2016a, b; Wang et al., 2015). However, OC and SOC annual mean concentrations, 3.5 ±1.5 µg m$^{-3}$ and 2.2 ±1.4 µg m$^{-3}$, respectively, were higher than those from Mt. Yulong. Therefore, carbonaceous matter associated with OC (e.g., SOC, POC) in the GHZ basin was mainly contributed from fossil fuel

(vehicle) emissions. The GHZ basin has local emissions from the frequent and intense tourism activities, including the use of vehicles, which is totally different from that of Mt. Yulong. Maximum seasonal mean BC and OC concentration occurred in the winter season (BC: 2.42 ±0.74 µg m$^{-3}$, OC: 4.09 ±1.7 µg m$^{-3}$, respectively) in the GHZ basin, and their lowest concentrations conventionally occurred in monsoons season (BC: 0.75 ±0.3 µg m$^{-3}$, OC: 2.8 ±0.8 µg m$^{-3}$, respectively). The pre-monsoon





season had the highest SOC concentration of 2.45 $\pm$ 1.3 µg m$^{-3}$, ranging from 0.17-6.1 µg m$^{-3}$. The mean SOC concentration in the monsoon season was larger than the post-monsoon SOC in the GHZ basin (Table 2), which was typically due to the extensive vehicle emissions and strong photochemical reactions during the monsoon

season. The seasonal variation of carbonaceous aerosols found in the GHZ basin was somewhat different from that found in Mt. Yulong, probably because of the distinct elevation difference and different degrees of human activities between the Mt. Yulong and the GHZ sampling sites.

The OC/BC ratios at the two sites were relatively low, and they have distinct

seasonal variations and spatial differences over the Mt. Yulong region. The annual average value was 2.06 $\pm$ 3.38 for Mt. Yulong, with the highest value occurred in the pre-monsoon season (3.67 $\pm$ 5.7) and the lowest value in the post-monsoon season (0.79 $\pm$ 0.4) (Table 1). Monthly variation of the average OC/BC ratio was determined by the relative concentrations of BC and OC in aerosol, for example, the lowest

OC/BC ratio occurred in post-monsoon was due to substantially high BC concentrations in that season (Fig. 6). The annual mean OC/BC ratio in samples from the GHZ basin was 2.9 $\pm$ 1.8, while the monsoon season had the highest value (4.4 $\pm$ 2.0) and the winter season had the lowest value (1.93 $\pm$ 1.58) (Table 2). Previous studies suggested that OC/BC ratios from biofuel and biomass burning emissions are

generally higher than those from fossil fuel combustion (Cao et al., 2013; Ram et al., 2012; Cong et al., 2015a; Wan et al., 2017). Strong photochemical reactions and tourism activities in the monsoon season were likely the main factors that result in relatively high OC/BC ratios in GHZ.

3.2 Optical properties of BC

The corrected MAE values of BC at 632 nm were 7.38 $\pm$ 1.01 and 6.25 $\pm$ 0.46 m$^{2}$ g$^{-1}$ for Mt. Yulong and GHZ samples, respectively. The BC MAE has distinct seasonal variations, with the peak of BC MAE values in the pre-monsoon and monsoon seasons at Mt. Yulong and GHZ, respectively (Fig. 7). The high MAE values suggest an enhancement of MAE (or absorption amplification) by external-coating with OC

(particularly SOC) (Cheng et al., 2011a; Knox et al., 2009; Schnaiter et al., 2005).



Strong seasonal and spatial differences of BC MAE values in Mt. Yulong and GHZ largely related to OC and SOC abundance. Furthermore, the variations of weekly mean MAE are consistent with the variations of OC concentration and SOC/OC ratio in Mt. Yulong (Fig. 8 and 9), as well as agree well with the trend of OC/BC ratio (Fig.

9). MAE and POC/OC ratio, ATN, and $BC_s$ are negatively correlated. The correlations between MAE and OC/BC ratio, SOC/OC ratio, and OC appear to be substantially influenced by the OC abundance. The availability of OC or SOA for external-coating is somewhat responsible for the variations of mean BC MAE (Knox et al., 2009). Values of atmospheric BC MAE are also dependent on the extent of

internal-mixing of the BC with other substances (Cappa et al., 2012; Schnaiter et al., 2005). Atmospheric BC light absorption is linearly proportional to the BC concentration since BC particles are small enough (Schwarz et al., 2013).

Many previous studies have quantified the BC MAE values at various sites (Bond and Bergstrom, 2006; Cheng et al., 2011a; Knox et al., 2009; Ram and Sarin, 2009; Li

et al., 2016c). However, large uncertainties exist among different calculation approaches. Measurement methods of ATN and $EC_s$ (various temperature protocols) definitely affect the BC MAE (Cheng et al., 2011a, b; Li et al., 2016c). In addition, biomass burning and brown carbon (BrC) decrease the BC MAE. BrC is less absorptive comparing with pure BC (Cheng et al., 2011a). BrC emitted from biomass

burning considerably lower MAE values (Jeong et al., 2004). Whereas coating by organic aerosol (particularly SOC) or mix-state can enhance the MAC values (Knox et al., 2009).

3.3 Controls of carbonaceous matter components

Our results show that carbonaceous matter (BC, OC, SOC) in aerosol exhibited an

apparent spatial discrepancy between Mt. Yulong and GHZ. In addition to the elevation difference, other potential factors accounting for this discrepancy need to be evaluated in depth. Concentrations of carbonaceous aerosols from GHZ were higher than those from Mt. Yulong, except for BC concentration, which to a large extent, indicate that carbonaceous matter in the Mt. Yulong region was primarily due to local

vehicle (tour bus) emissions (e.g., 1.87 Tg C in 2016 in Yunnan province, China).





In addition, inter-annual differences of carbonaceous aerosol from Mt. Yulong and GHZ were also distinct (Fig. 10). The annual mean concentrations of carbonaceous matter collected in 2016 were lower than those in 2015 for the two sites (Fig 10 a, b), indicating that the improvement of local atmospheric environment in the Mt. Yulong

region might ascribe to strict mitigation measures. For example, the amount of soot emissions (2.44 Tg C) in 2016 were reduced by 21.76% compared to 2015 in Yunnan province (http://www.zhb.gov.cn/). The average OC/BC ratios also decreased in 2016 compared to 2015, whereas the mean SOC/OC and POC/OC ratios have no obvious difference between the two years in both Mt. Yulong and GHZ (Fig. 10). However, it

is complex to accurately evaluate the inter-annual variations of OC/BC ratios in the current study.

We also compare atmospheric BC concentrations in Mt. Yulong with other interested areas. It shows that BC concentrations in Mt. Yulong aerosol were relatively low and, among the compared values, most of BC concentrations were typically

within the range of 1.0-3.0 μg m$^{-3}$ (Fig. 11), while some of values were extremely low (close to 0.5 μg m$^{-3}$) and high (above 6.0 μg m$^{-3}$). The low BC concentrations were typically found in the TP, e.g., QSMS, TP (Cong et al., 2015a), pre-monsoon season in Mt Yulong. Whereas the BC concentrations in Agra, India (6.1 ±0.83 g m$^{-3}$) (Pachauri et al., 2013) were almost three folds of values found in Mt. Yulong. Agra and Lumbini

have been identified as regions in the world that are highly affected by biomass burning (Wan et al., 2017). A large amount of carbonaceous aerosols emitted from those regions can reach Mt. Yulong by crossing the Himalayas (e.g., Lüthi et al., 2014), which substantially influence the Mt. Yulong region.

3.4 Source apportionments of carbonaceous aerosols

Aerosol vertical distributions from CALIOP retrievals often reveal that smoke plume could reach approximately 6 km (Fig. 12), which is higher than most of the mountains and mountain glaciers in the Himalaya regions. Some typical CALIOP/CALIPSO transections, having strong backscattering signal (i.e., 532 nm total attenuated backscatter), show spatially continuous atmospheric pollutant layers

from the ocean all the way to Mt. Yulong (Fig. 12), indicating a penetration of smoke



plume toward the inland of the TP. It was reported that the pre-monsoon season is the major vegetation-fire period in the foothill areas of the southern Himalaya (Vadrevu et al., 2012; Putero et al., 2014), and the winds surrounding the Himalayas and TP could facilitate the transport of carbonaceous matter from South Asia to the Himalayas

(Cong et al., 2015b; Dong et al., 2017b).

We analyze the CAM5 model results to quantify the source attributions of BC at the surface measurement stations. BC emissions from each of the four source regions in the surrounding area (i.e., South Asia, East Asia, Middle East and Southeast Asia) are explicitly tracked. Figure 13 shows the annual and seasonal mean relative

contributions from the tagged source regions. The two sampling sites locate in the same model grid box, as marked in the figure. The modeled near-surface BC is predominately (more than 90%) from South Asia and East Asia. East Asia has a dominant contribution in the monsoon, post-monsoon and winter seasons, while South Asia dominates in the pre-monsoon season (Table 3). Thus relatively high BC

concentrations in the high atmosphere (approximately 5000 m asl) of Mt. Yulong indicate an important role of the long-range transport of biomass and biofuel burning emissions from South Asia (e.g., Cong et al., 2015a, b; Lee et al., 2013; Li et al., 2016 a, b; Lüthi et al., 2014) and East Asia. As discussed by Wang et al. (2015), circulation patterns during the monsoon and non-monsoon seasons largely determine the seasonal

variations in the transport of aerosols from the different major sources to the southeastern TP. Strong precipitation during the monsoon season can substantially remove atmospheric BC during the transport, especially, from South Asia. Although smoke plumes can sometimes be lifted over the natural block of the Himalayas, they have relatively less important contribution to the surface than to the upper-level BC

concentrations. Therefore, emissions from East Asia, including location sources, have a dominant contribution to the near-surface BC at the Mt. Yulong sites during the monsoon and post-monsoon seasons, as well as the winter season. The seasonal changes in source apportionments also have an implication on the cause of variations in OC/BC ratios over the southeastern TP (e.g., Wang et al., 2015).

**4. Conclusions and remarks**





Carbonaceous aerosols from the Mt. Yulong region and GHZ basin were measured to investigate the spatio-temporal variations and light-absorbing properties. Results of first two years of continuous observations show that the annual mean BC and OC concentrations in aerosol were $1.51 \pm 0.93$ and $2.57 \pm 1.32$ μg m$^{-3}$, respectively. The average SOC concentrations were $1.67 \pm 1.15$ μg m$^{-3}$ determined using BC-tracer method.

Concentrations of carbonaceous matter displayed distinct seasonal differences, with the lowest content found in monsoon season and the highest concentration in winter season. Monthly mean BC concentrations in aerosols from monsoon to post-monsoon season were higher than OC, a large impact from biomass burning emissions in the high atmosphere (approximately 5000 m asl) of Mt. Yulong. The seasonal differences of carbonaceous matter found in GHZ basin were different with that in Mt. Yulong, distinct elevation difference and different degrees of human activities between the two sites were the main reasons account for the discrepancy. Furthermore, high carbonaceous matter associated with OC (e.g., SOC, POC) in carbonaceous aerosols in GHZ basin was mainly contributed from vehicle emissions. Therefore there have distinct spatial differences in the concentrations of carbonaceous matter in this glacierized region. Moreover, inter-annual differences of carbonaceous aerosols in Mt. Yulong and GHZ were also distinct. The annual mean concentrations of carbonaceous matter in 2016 were totally lower than those in 2015, indicating the improvement of local atmospheric environment in the Mt. Yulong region.

The annual mean OC/BC ratio was $2.45 \pm 1.96$ in Mt. Yulong, with the highest value in monsoon season ($4.4 \pm 2.0$) and the lowest in winter ($0.79 \pm 0.4$). Strong photochemical reactions and local tourism activities in monsoon season were the main factors resulting in relatively high OC/BC ratios in the Mt. Yulong region, particularly in GHZ basin.

BC MAE was quantified using a thermal-optical carbon analyzer, and was measured at 632 nm under the quartz filter-based method. The corrected mean BC MAE at 632 nm was $6.82 \pm 0.73$ m$^2$ g$^{-1}$ in the Mt. Yulong region, comparable with the results in other studies. The trends of weekly mean BC MAE consistent with the



variations of OC concentrations, SOC/OC, and OC/BC ratios. Strong seasonal and spatial discrepancies of BC MAE in the study area were largely related to the OC and SOA abundance. The enhancement of MAE was mainly due to external-coating of OC (particularly SOA) and/or mixing state.

To quantitatively estimate the source apportionment of BC at the Mt. Yulong sites, we used a global aerosol-climate model, in which BC emissions from four regions (i.e., South Asia, East Asia, Middle East and Southeast Asia) are explicitly tracked. The five-year (2010-2014) mean results show that East Asia has the largest contribution (52%) to the annual mean near-surface BC concentration at the two sites,

followed by South Asia (43%). There is a quite strong seasonal variation in the source apportionment. East Asia has a dominant contribution in the monsoon and post-monsoon seasons, while South Asia dominates in the pre-monsoon season.

**The authors declare that they have no conflict of interest.**

**Acknowledgements**

This work was supported by the National Natural Science Foundation of China (41601071, 41421061, 41225002), the Chinese Academy of Sciences (CAS) (KJZD-EW-G03-04), the "Light of West China" Program (Y62992) and the

independent program of SKLCS (SKLCS-22-2017), and China Postdoctoral Science Foundation (2015M582725, 2016T90963) and Youth Talent Program of CAREERI CAS (Y551C11001). R. Zhang acknowledges support from NSFC (41605041), Jiangsu Provincial Science Fund (BK20160621), Fundamental Research Funds for the Central Universities (020714380020) and International Postdoctoral Exchange

Fellowship (20160046). H. Wang and Y. Qian acknowledge support from the U.S. Department of Energy (DOE), Office of Science, Biological and Environmental Research as part of the Earth System Modeling program. The Pacific Northwest National Laboratory (PNNL) is operated for DOE by Battelle Memorial Institute under contract DE-AC05-76RLO1830. The model simulations were performed using

PNNL Institutional Computing resources.



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




Table 1

Statistical results of BC, OC, SOC, and POC concentrations (µg m$^{-3}$) and OC/BC ratios in aerosol from Mt. Yulong during Dec. 2014-Dec. 2016.

| | Annual (n=120) | | Winter (n=36) | | Pre-monsoon (n=34) | | Monsoon (n=33) | | Post-monsoon (n=17) | |
|---|---|---|---|---|---|---|---|---|---|---|
| | Mean±SD | Range | Mean±SD | Range | Mean±SD | Range | Mean±SD | Range | Mean±SD | Range |
| OC | 1.65±1.14 | 0.07-5.96 | 1.75±0.8 | 0.51-3.66 | 1.37±1.2 | 0.07-5.73 | 1.34±0.9 | 0.24-4.6 | 2.57±1.3 | 0.99-5.96 |
| BC | 1.53±1.49 | 0.02-6.83 | 1.81±1.5 | 0.02-6.83 | 0.55±0.4 | 0.03-1.77 | 1.25±1.2 | 0.04-3.9 | 3.51±1.2 | 1.22-5.8 |
| OC/BC | 2.06±3.38 | 0.35-20.9 | 1.45±1.1 | 0.38-5.55 | 3.67±5.7 | 0.7-22.9 | 1.85±1.8 | 0.42-8.3 | 0.79±0.4 | 0.35-2.27 |
| SOC | 1.15±0.96 | 0.06-5.44 | 1.15±0.5 | 0.32-2.56 | 1.18±1.2 | 0.06-5.44 | 0.96±0.8 | 0.19-4.5 | 1.42±1.2 | 0.19-4.59 |
| POC | 0.53±0.53 | 0.009-2.4 | 0.63±0.5 | 0.008-2.39 | 0.19±0.1 | 0.01-0.62 | 0.43±0.4 | 0.01-1.3 | 1.24±0.4 | 0.42-2.04 |





Table 2

Statistical results of BC, OC, SOC, and POC concentrations (µg m$^{-3}$) and OC/BC ratios in aerosol from GHZ during Dec. 2014-Dec. 2016.

| | Annual (n=116) | | Winter (n=27) | | Pre-monsoon (n=47) | | Monsoon (n=28) | | Post-monsoon (n=12) | |
|---|---|---|---|---|---|---|---|---|---|---|
| | Mean±SD | Range | Mean±SD | Range | Mean±SD | Range | Mean±SD | Range | Mean±SD | Range |
| OC | 3.5±1.5 | 1.1-10 | 4.09±1.7 | 2.3-9.9 | 3.7±1.7 | 1.3-10 | 2.8±0.8 | 1.1-4.7 | 3.1±1.1 | 2.0-5.6 |
| BC | 1.5±0.9 | 0.15-4.5 | 2.42±0.74 | 1.1-4.2 | 1.4±0.7 | 0.28-4.5 | 0.75±0.3 | 0.15-1.5 | 1.58±0.9 | 0.63-3.2 |
| OC/BC | 2.9±1.8 | 0.87-8.8 | 1.93±1.58 | 0.87-8.9 | 2.72±1.3 | 1.1-8.3 | 4.4±2.0 | 1.7-8.5 | 2.36±0.9 | 1.1-3.9 |
| SOC | 2.2±1.4 | 0.16-8.9 | 2.06±1.9 | 0.47-8.9 | 2.45±1.3 | 0.17-6.1 | 2.14±0.7 | 0.9-3.8 | 1.67±0.8 | 0.68-3.4 |
| POC | 1.3±0.8 | 0.13-4 | 2.11±0.7 | 0.9-3.7 | 1.25±0.6 | 0.25-4.0 | 0.67±0.3 | 0.13-1.3 | 1.4±0.8 | 0.56-2.9 |





Table 3 Fractional contribution (%) from four major source regions including south Asia (SAS), East Asia (EAS), Southeast Asia (SEA), and Middle East (MDE) to BC surface concentrations over the Mt. Yulong region in winter (December-February), pre-monsoon (March-May), monsoon (June-September), post-monsoon (October-November), and all months during the model simulation time period (2010-2014).

| Source region | Winter | Pre-monsoon | Monsoon | Post-monsoon | Annual |
|---|---|---|---|---|---|
| SAS | 40.8 | 67.63 | 17.12 | 27.14 | 42.49 |
| EAS | 53.76 | 25.37 | 77.96 | 68 | 52.1 |
| SEA | 3.03 | 2.52 | 3.73 | 3.18 | 3.02 |
| MDE | 1.06 | 2.39 | 0.47 | 0.67 | 1.02 |



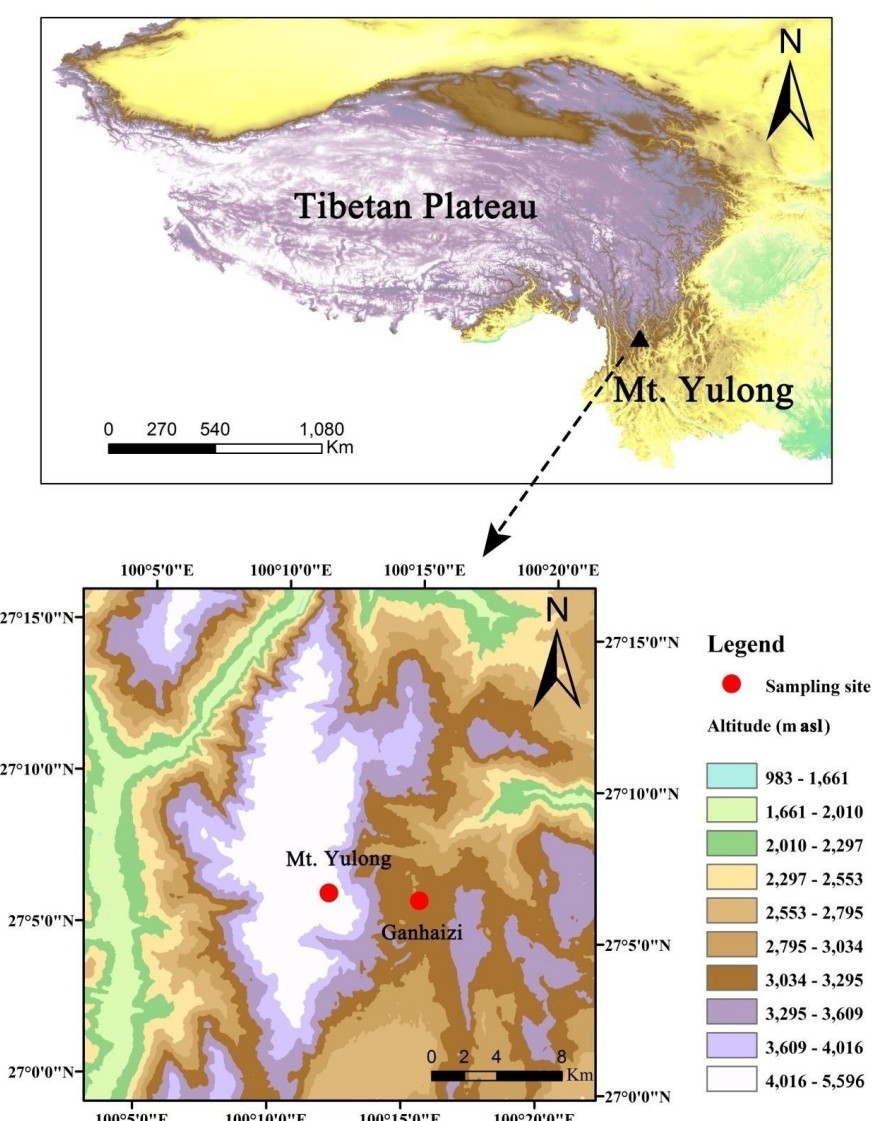

Fig. 1 Location of Mt. Yulong in the Tibetan Plateau and the sampling sites at Mt.
Yulong and Ganhaizi basin.



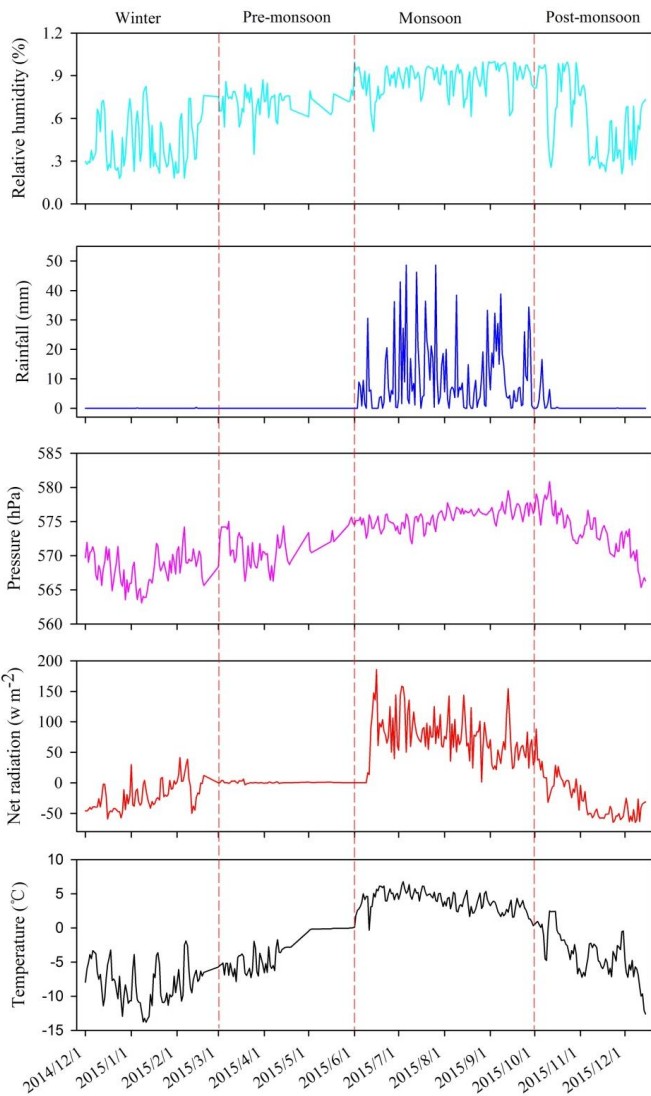

Fig. 2 Time series of meteorological parameters (air-temperature, net radiation, pressure, rainfall, relative humidity) at Mt. Yulong from December 2014 to December 2015. Monsoon and non-monsoon seasons (including winter, pre-monsoon, post-monsoon seasons) are divided by vertical lines.



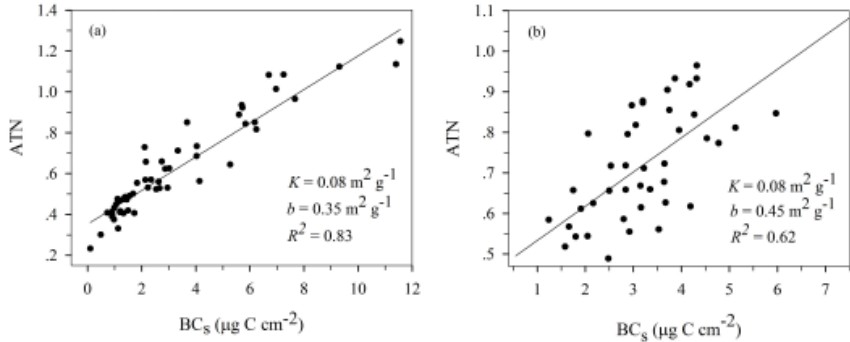

Fig. 3 Dependence of optical attenuation (ATN) detected at 632 nm on the BC loading (BC$_s$) for (a) the Mt. Yulong sampling site and (b) the GHZ sampling site. Results of linear regression are displayed with $K$ as the slope and $b$ as the intercept.




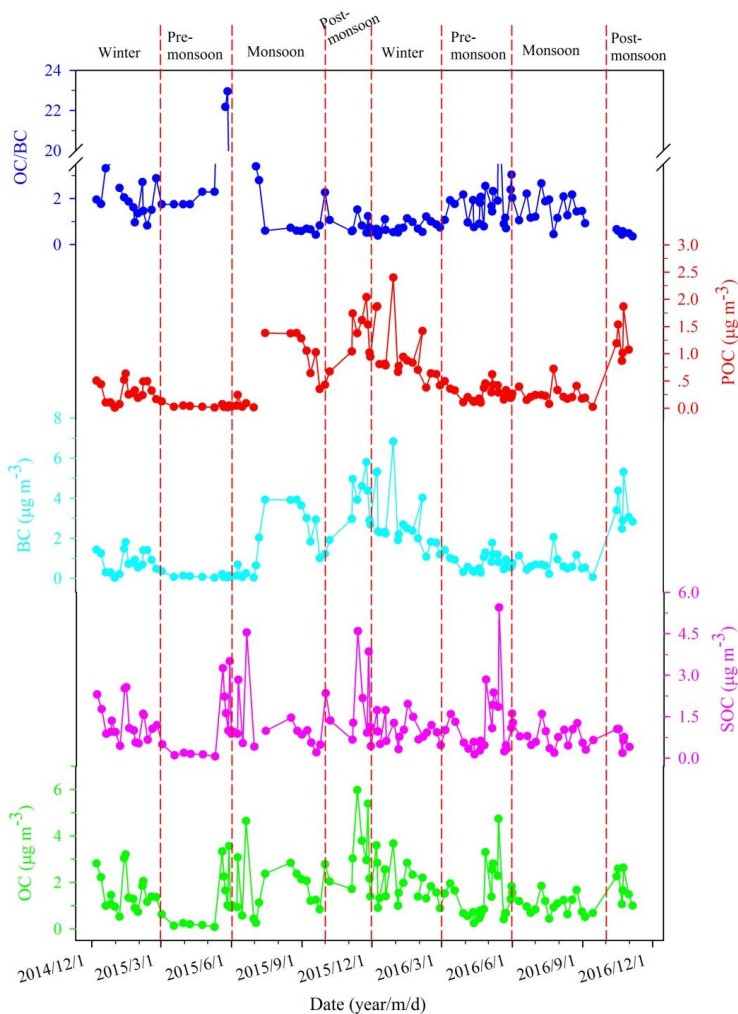

Fig. 4 Seasonal variations of BC, OC, SOC, and POC concentrations and OC/BC ratio

from Mt. Yulong during Dec. 2014-Dec. 2016.





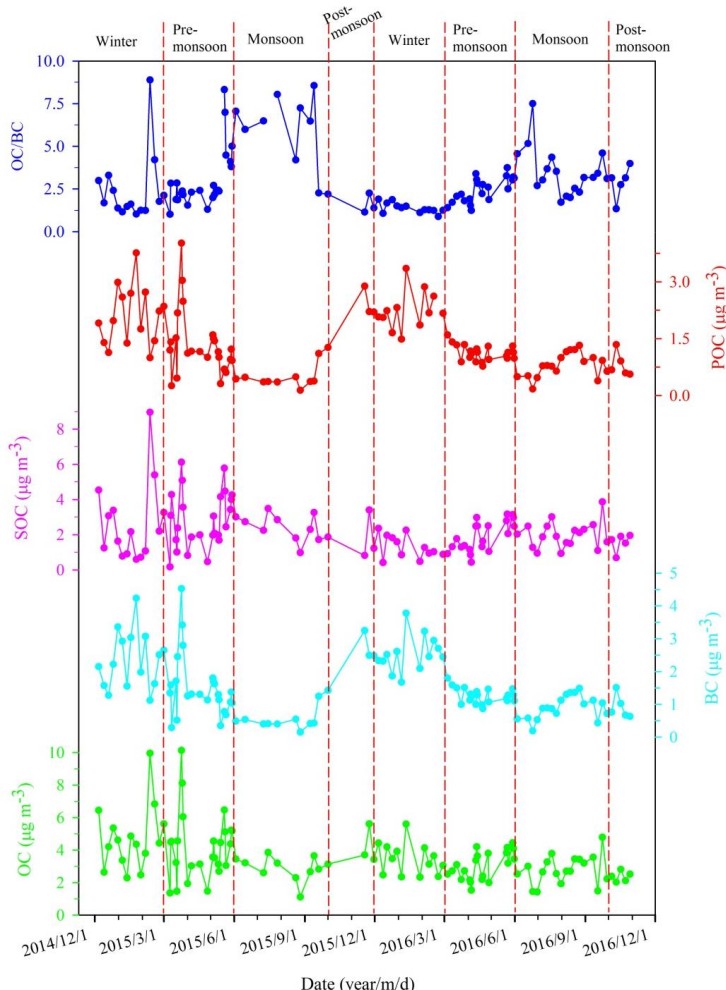

Fig. 5 Seasonal variations of BC, OC, SOC, and POC concentrations and OC/BC ratio

from GHZ basin during Dec. 2014-Dec. 2016.



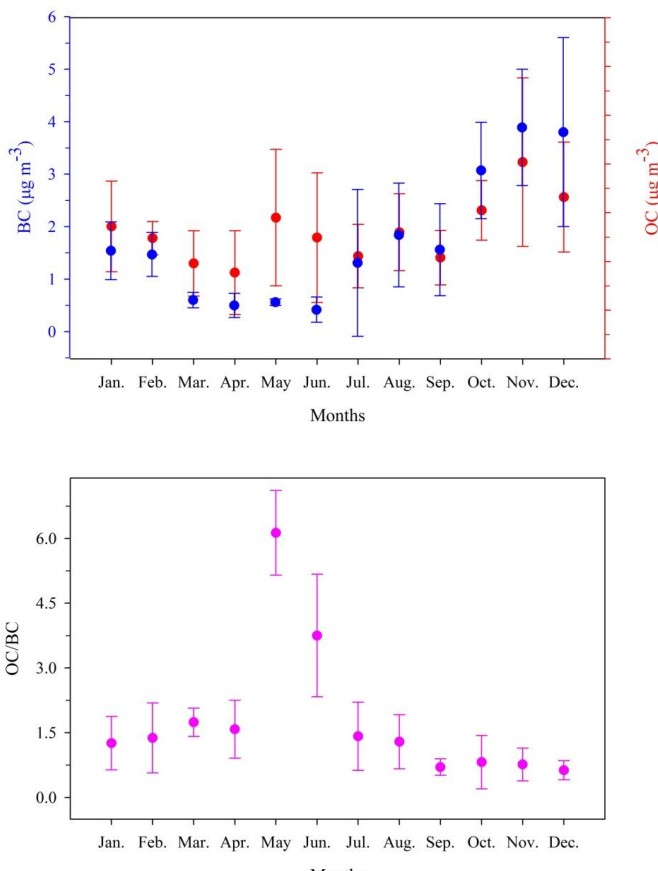

Fig. 6 Monthly averaged BC, OC concentrations and OC/BC ratios from Mt. Yulong.





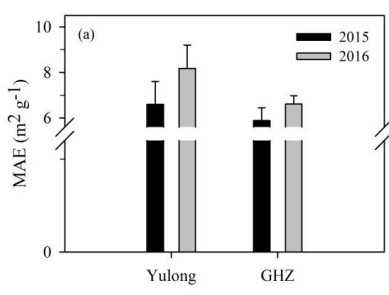 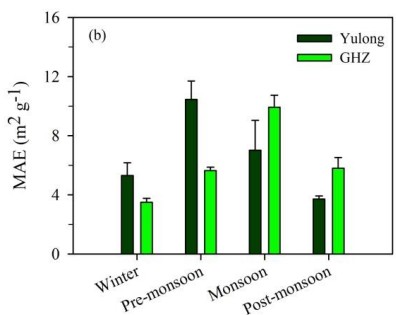

Fig. 7 Histogram of (a) annual mean and (b) seasonal mean MAE values of atmospheric BC from Mt. Yulong and GHZ sampling sites during December 2014-December 2016.



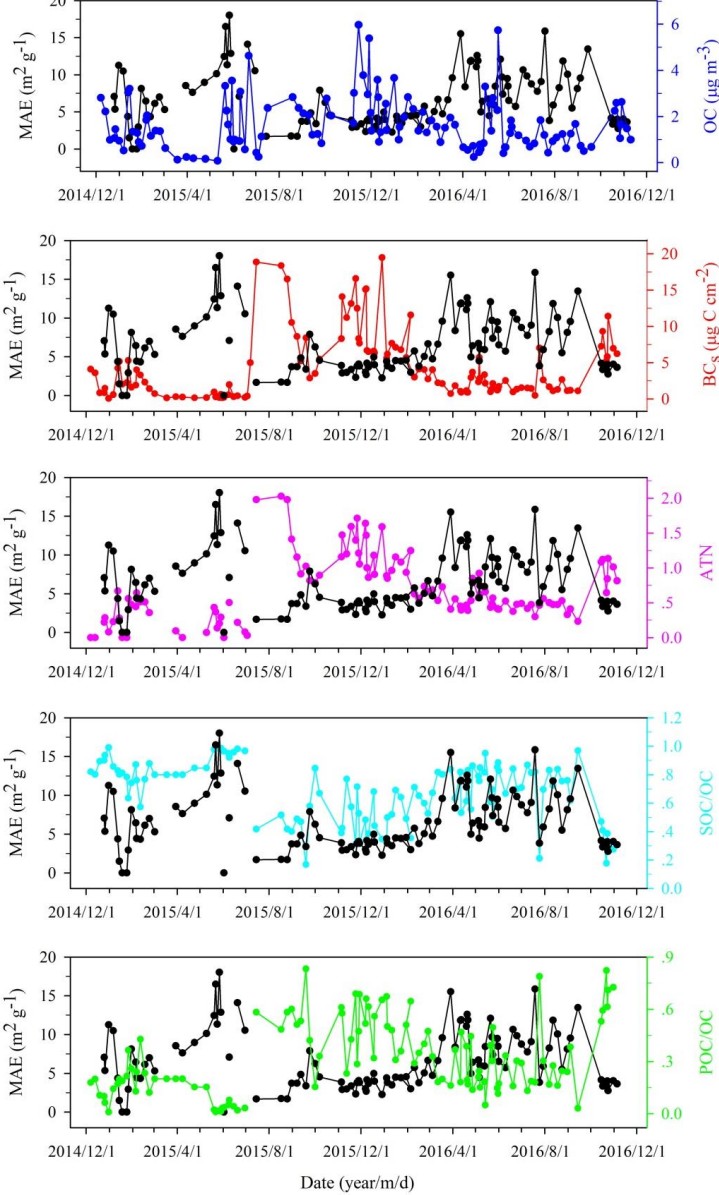

Fig. 8 Weekly mean MAE values in Mt. Yulong aerosol during December 2014 to December 2016. The variations of POC/OC and SOC/OC ratios, ATN, BC$_s$, and OC concentrations are also displayed for comparison with MAE.





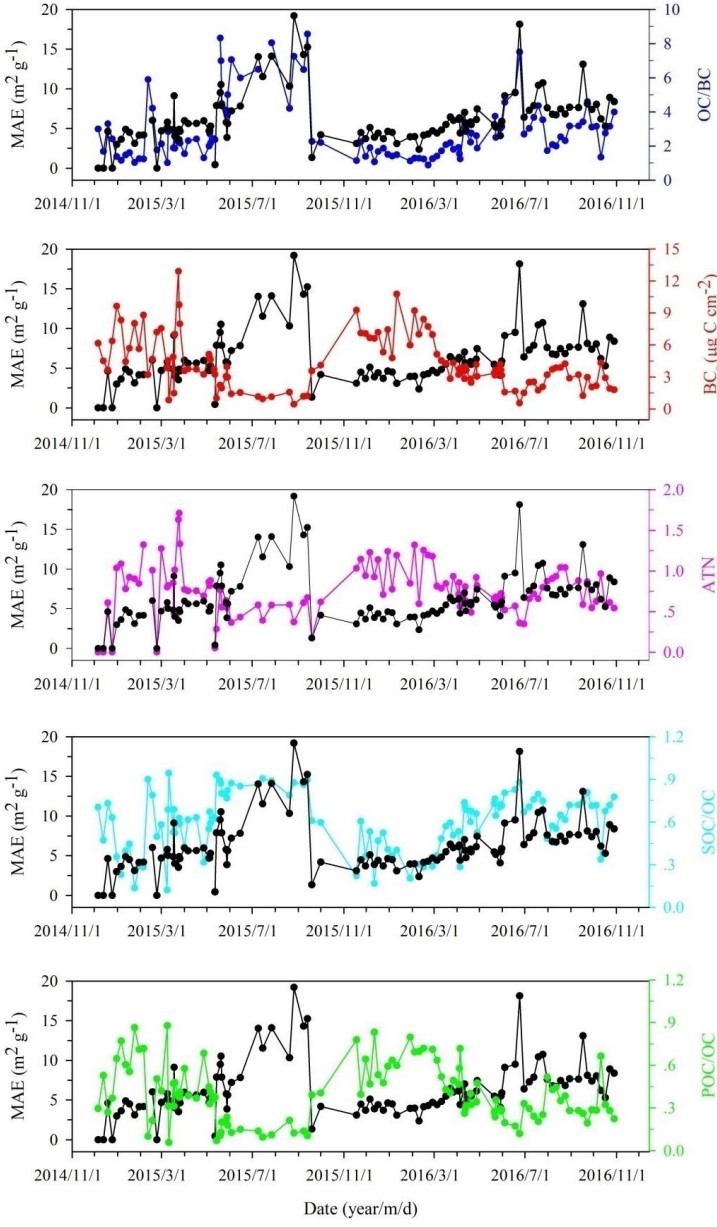

Fig. 9 Weekly mean MAE values in GHZ aerosol during December 2014 to December
2016. The variations of POC/OC, SOC/OC, and OC/BC ratios, ATN, $BC_s$ values are
displayed for comparison with MAE.





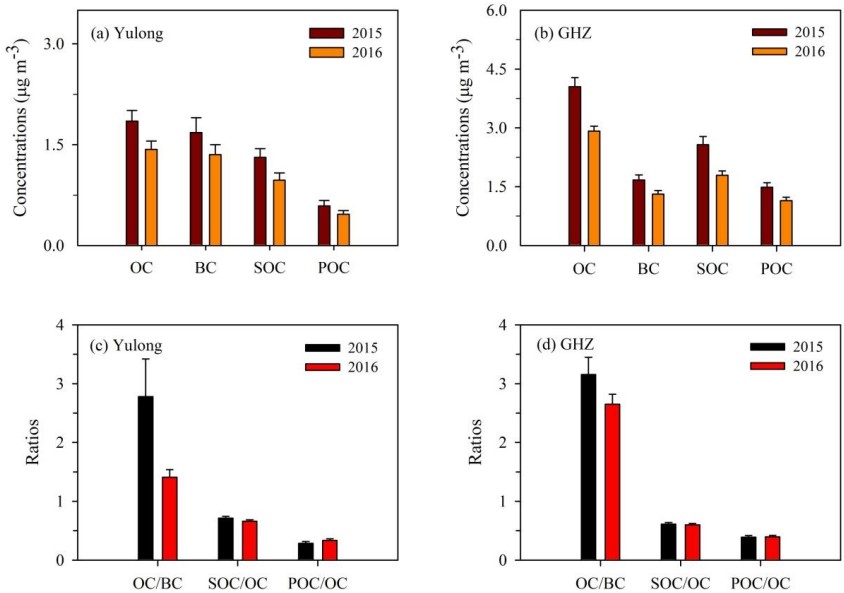

Fig. 10 Inter-annual differences of OC, BC, SOC, and POC concentrations at (a) Yulong and (b) GHZ site. The OC/BC, SOC/OC, and POC/OC ratios are shown in (c, d).





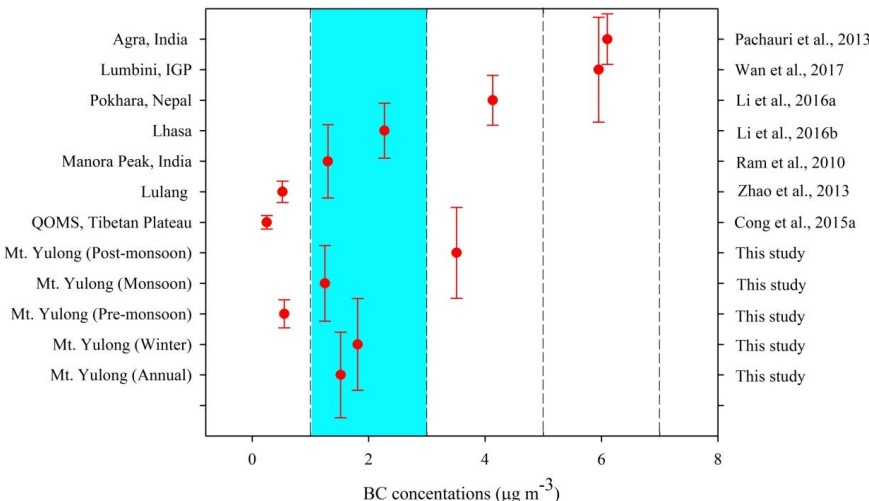

Fig. 11 Comparison of BC concentrations from Mt. Yulong and other areas of interest.





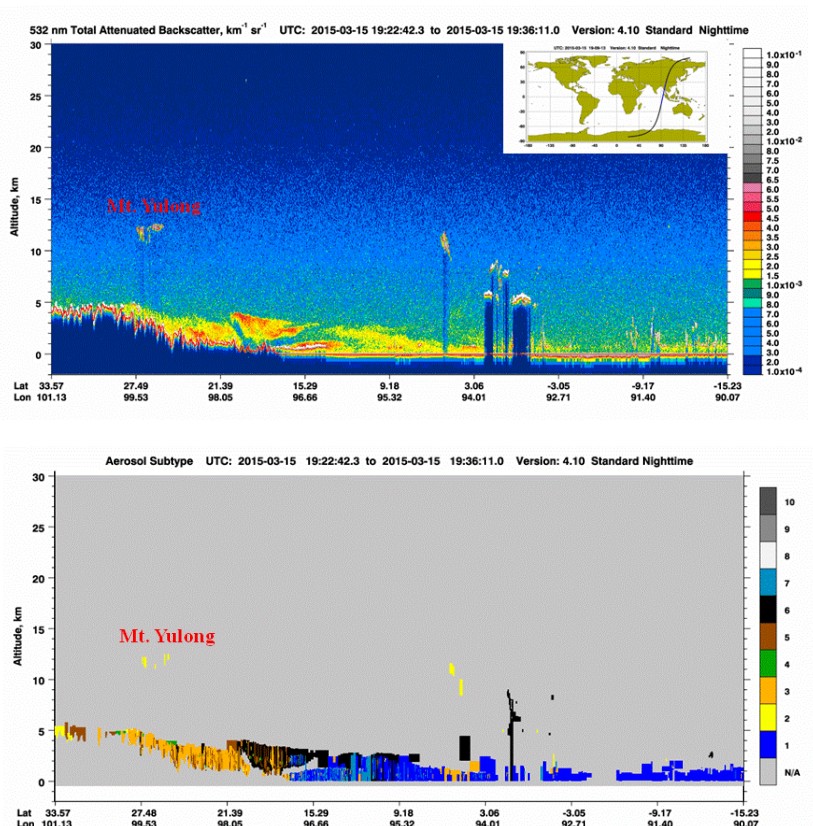

Fig. 12 CALIPSO retrieved backscatter signal at 532 nm (upper panel) and aerosol
sub-type information (bottom panel) on 15 March, 2015. The Mt. Yulong region was
covered by a thick aerosol layer (mainly consisting of polluted smoke and dust) in the
high atmosphere (above 6000 m asl), likely transported far from their source regions.
CALIPSO profiles were obtained from (http://www-calipso.larc.nasa.gov). The
CALIOP-derived reflectivity is usually taken as an indicator to reflect the structure of
atmospheric layers since it dependents on mass concentration and optical properties of
atmospheric aerosol (Bou et al., 2010; Dong et al., 2017a). The topography is outlined
by a solid red-line. Suspended dust and aerosol pollutants are in orange and red.



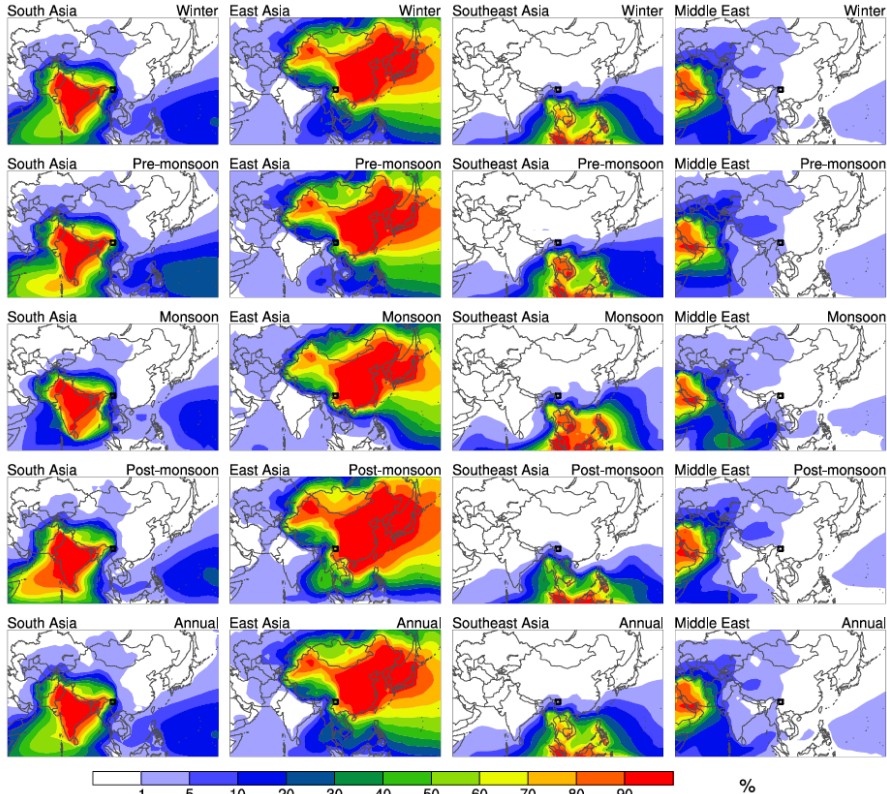

Fig. 13 Annual and seasonal (winter, pre-monsoon, monsoon and post-monsoon) mean relative contributions of emissions in four tagged source regions, including South Asia, East Asia, Southeast Asia and Middle East, to near-surface BC concentrations. The black box in each panel marks the grid box where the Yulong and GHZ sites locate.