# Peer review of "Seasonal variation and light absorption property of carbonaceous aerosol in a typical glacier region of the southeastern Tibetan Plateau"

_Atmospheric Chemistry and Physics, 2017_

## Referee Comment (RC1) · Anonymous Referee #2 · 22 Dec 2017

This paper reports two-years filter-based measurements of carbonaceous aerosols at two sites a typical glacierization region of the Tibetan Plateau. The dataset provided by this manuscript is important because of the unique location of the experimental site and the profound implication of carbonaceous aerosols deposition on glacier melting. However, I found that some of the conclusions were not solid and lack of sufficient data analysis (or at least relevant reference) support; many of the discussions were superficial and need to be revised. I suggest this manuscript may be accepted for a publication in ACP after the authors could address the following comments . Major concerns: 1. Despite Mt. Yulong and Ganhaizi are special sampling sites, I have not seen much interesting or unique scientific findings. Except providing the basic dataset (numerous

descriptive words like "high", "low", "important"), neither was the dataset analyzed and discussed enough, nor were the conclusions obtained in a solid way. Many statements lack sound evidence or support. Here are some examples: P8, line 2-4; P10, line 17-20; P10, line 28-30; P11, line 13-15; line 23-25; P12, line 2-8; line 21-23; P13, line 27-30; P14, line 2-5; P15, line 25-29. 2. The authors claim that the sampling sites are in a typical glacier region of the Tibetan Plateau; that is not true. The two sampling sites are very close to each other: The Mt. Yulong site is likely a mountaintop site and the Ganhaizi site is located on the foot of Mt. Yulong. In this regard, the title of this MS is apparently overstated because two ground-based measurement sites are not enough to capture the characteristics of "spatio-temporal variability of carbonaceous aerosol". Two sites are "...separated from the urban area, but is an increasingly popular tourist destination, with a geological museum and a golf course. Moreover, there is a spacious parking lot and a tourist dining-center in GHZ." Local sources may be important as already stated in the text. 3. I appreciate the efforts of a global aerosol-climate model used in the study to quantify the source attribution of BC. However, the model simulation didn't cover the sampling period of this study, which dramatically diminish the reliability and importance of their modeling work. The authors found that there was a significant inter-annual variation of carbonaceous aerosols induced by emission reduction, I don't think they a climate model with outdated emission input can be used to infer the sources of carbonaceous aerosols measured in the current work. 4. The current form of the abstract is informative but frequently distract my focus. Please show the readers your most important and exciting findings.

Specific comments: 1. Title: overstated as pointed out above. 2. P3, line 8: missing reference. 3. P4, line 1: specify the "absorbing aerosols". 4. Line 7: partially correct. 5. Line 13: overstated. 6. Line 17: For BC particles, 7. P6, line 27: specify the number of samples collected at the two sites. 8. P7, line 3-5: this is nonsense if you have no attempt to give more details. 9. Line 9: specify the temperature protocol you adopted. 10. Line 20: the authors suggested different values of (OC/BC)min should be used to estimate SOC for different dataset (pre- or after-monsoon season), but I didn't see the

discussion about data processing. 11. P8, line 1-4: given that "there is no standard or more perfect method of OCsec estimation (Yu et al., 2007)", how can you claim that "BC-tracer method employed here is reliable in determining SOC concentrations." 12. P10, line 29: needs solid evidence. 13. P11, line 9: NCO-P? 14. Line 13-15: needs solid evidence. 15. Line 17: keep a constant significance digit. 16. Line 17-20: lacking of logic. 17. Line 21: the annual average concentrations of OC and SOC. 18. Line 23-25: any solid evidence? 19. P12, line3-5: missing reference or supporting evidence. 20. Line 6: "somewhat" should be avoided as much as possible in a scientific paper. 21. Line 5-8: needs solid evidence. 22. Line 21-26: needs solid evidence. 23. P13, line 18: any reference? 24. Line 25: "apparent spatial discrepancy"?, I don't think so. 25. Line 28-30: unconvinced evidence. Any reference? 26. P14, line 2-5: I suggest the authors to explore the effects of precipitation on the difference of carbonaceous aerosols mass loadings measured in different years. 27. Line 12-16: I am not sure if these results in previous studies were derived from TSP samples. 28. P15, line 25-27: needs more solid evidences.

---

## Referee Comment (RC2) · Anonymous Referee #1 · 26 Dec 2017

Comments to the Author

This manuscript presented a topic with spatial and temporal variability and light absorption property of carbonaceous aerosols (OC and BC) in a typical glacierization region of Tibetan plateau. However, there are several key issues that authors should address and enhance/clarify (see below):

a) The annual mean OC/BC ratio was found to be highest in the monsoon season and lowest in winter. The author needs to mention, which types of photochemical reactions are involved that increased OC/ BC ratio. Generally, photochemical reaction changing their physical and chemical properties from the original molecule of the substance. When this occurs, these molecules tend to form a new structure, or combine with each other or other molecules. This may change the state of OC or BC or OC/BC ratios.

b) Author mentioned that tourism activities in monsoon season were attributed to relatively high OC/BC ratios in Mt. Yulong region, especially in GHZ basin. Firstly,author did not define tourism activities in whole manuscript. Secondly, how do author think "tourism activities" term used for vehicles or buses emission. Please explain in scientific literature that indicates tourism related activities contributed to OC, BC emissions with references to justify the statement.

c) In the result and discussion part "human activities" creating difference in Mt. Yulong and GHZ basin. This is a major observation indicated by author, but there is no concrete evidence available to support the results particularly what types of human activities?

d) Authors stated that "A large amount of biomass burning emissions in the high atmosphere (around 500 m asl) in Mt. Yulong were probably long range transported from source regions". This observation is supported by the higher BC concentration than OC especially during monsoon season to post monsoon season. How do authors assume this observation without performing Trajectory analysis with HYSPLIT model? In order to find the transport of pollutant emission from biomass burning, author needs to perform trajectory analysis during the study period to support the results for Dec. 014 and Dec.2016.

e) Authors selected experiment dates (December 2014-Decemebr 2016) for time series of meteorological parameters at Mt. Yulong. On the other hand, in model experimental part for CAM5, selected dates are 2010 to 2014, although the author stated the reason but for actual results synchronization, it should be similar dates to give clearer picture and what is base to select these years?.

f) In result and discussion parts, author mentioned "seasonal changes in BC and OC sources (e.g., biomass burning vs. fossil fuel combustion) might play an important role for the variations of OC/BC ratios". How does seasonal change affect in OC/BC ratio? Explain the reason and factors involved in this change.

g) How did author's calibrate the Particulate sampling apparatus (TH150-A, Wuhan Tianhong INST Group? and the quality assurance of this apparatus is not explained in manuscript.

   Some minor revision is as fellows

a) In the abstract part line 2-3 write down the name for adjoining areas.
b) In the introduction part line 1-2 add Bond et al. 2007 reference to support the statement
c) In the introduction part line 3-4 use words as " human health and living species" instead of using "living organisms"
d) In the introduction part line 4 reference is missing.
e) Page 4; line 4 Tibetan plateau(TP), abbreviation is already used in abstract part, so need to use full word
f) Page 5; line 12 please elaborate the model names that are not considered absorption by OC with reference.
g) In material and method part page5; Line 29. This statement needs reference to support the arguments.
h) Page 13; use some other words instead of using "But"
i) Page 6; line 16, Is there any other pollution sources?, if yes please mention and explain their sources.
j) Page 9; line 20, write down "Then they" instead of using "They then"
k) Page 10; line 20 reference is missing.
l) Page 12; line 7 what type of human activities involved between Mt. Yulong and the GHZ sampling sites?
m) Page 14; line 17 write down the full name for "QSMS"
n) Page 17;line 11 add "in BC emission" after "contribution"
o) Page 41; Fig. 11. Is there any major difference or key observation that author found in current study which is actually different from others. Please explain in the footnotes of Fig.11

---

## Author Comment (AC1) · 5 Feb 2018

Response to referees' comments Journal: ACP Title: Seasonal variation and light absorption property of carbonaceous aerosol in a typical glacier region of the southeastern Tibetan Plateau Author(s): Hewen Niu et al. MS No.: acp-2017-865 MS Type: Research article Iteration: Correction Special Issue: Study of ozone, aerosols and radiation over the Tibetan Plateau (SOAR-TP) (ACP/AMT inter-journal SI)

We thank the referees for their careful review and insightful comments and suggestions, which greatly helped us to improve the manuscript. Below are our point-by-point responses and detailed corrections/revisions made to the manuscript accordingly.

[Figure]

RC 1 Anonymous Referee #2

This paper reports two-years filter-based measurements of carbonaceous aerosols at two sites a typical glacierization region of the Tibetan Plateau. The dataset provided by this manuscript is important because of the unique location of the experimental site and the profound implication of carbonaceous aerosols deposition on glacier melting. However, I found that some of the conclusions were not solid and lack of sufficient data analysis (or at least relevant reference) support; many of the discussions were superficial and need to be revised. I suggest this manuscript may be accepted for a publication in ACP after the authors could address the following comments. Major concerns: 1. Despite Mt. Yulong and Ganhaizi are special sampling sites, I have not seen much interesting or unique scientific findings. Except providing the basic dataset (numerous descriptive words like "high", "low", "important"), neither was the dataset analyzed and discussed enough, nor were the conclusions obtained in a solid way. Many statements lack sound evidence or support. Here are some examples: P8, line 2-4; P10, line 17-20; P10, line 28-30; P11, line 13-15; line 23-25; P12, line 2-8; line 21-23; P13, line 27-30; P14, line 2-5; P15, line 25-29.

Response: We have thoroughly revised the manuscript, particularly, to the places mentioned here, by providing solid evidence or support for each argument/statement. Detailed revisions to the paper for some arguments are discussed below in response to specific comments and we have now highlighted new findings of this work. After revision, we believe the manuscript is much improved and the interesting scientific findings are clear.

2. The authors claim that the sampling sites are in a typical glacier region of the Tibetan Plateau; that is not true. The two sampling sites are very close to each other: The Mt. Yulong site is likely a mountaintop site and the Ganhaizi site is located on the foot of Mt. Yulong. In this regard, the title of this MS is apparently overstated because two ground-based measurement sites are not enough to capture the characteristics of "spatio-temporal variability of carbonaceous aerosol". Two sites are "... separated

from the urban area, but is an increasingly popular tourist destination, with a geological museum and a golf course. Moreover, there is a spacious parking lot and a tourist dining-center in GHZ." Local sources may be important as already stated in the text.

Response: We agree with the referee that the two sampling sites are close to each other and cannot be used to characterize "spatial variability". However, there is an obvious elevation difference between the two sampling sites. The sampling site in Mt. Yulong (4510 m asl) is above the terminal of the Baishui Glacier (4430 m asl), while GHZ (3054 m asl) is a flat basin and nearly at the foot of Baishui Glacier. Both of the two sampling sites are in glacier region. We have revised the title of this manuscript accordingly to "Seasonal variation and light absorption property of carbonaceous aerosol in a typical glacier region of the southeastern Tibetan Plateau".

3. I appreciate the efforts of a global aerosol-climate model used in the study to quantify the source attribution of BC. However, the model simulation didn't cover the sampling period of this study, which dramatically diminish the reliability and importance of their modeling work. The authors found that there was a significant inter-annual variation of carbonaceous aerosols induced by emission reduction, I don't think they a climate model with outdated emission input can be used to infer the sources of carbonaceous aerosols measured in the current work.

Response: We agree that it would be more desirable to have the model simulation cover the same time as the sampling period for observations. However, as we mentioned in the manuscript, the emissions are only available up to year 2014. Even if we had an up-to-date global emission inventory available, which usually doesn't have a good estimate of small-scale local sources, a coarse-resolution climate model wouldn't be able to capture the quantitative source attributions of the measured BC at the sampling sites. Therefore, the model experiment was not designed to simulate the two-year observations of aerosols in the sampling area, but rather for a recent time period (2010-2014) to estimate the climatological mean source attributions and seasonal variations of near-surface BC concentrations, which has approved to be useful to help understand the observational analysis in the TP region (e.g., Wang et al., 2015). We did find a significant difference in measured carbonaceous aerosols between 2015 and 2016, which was partly attributed to emission changes based on some emission data of Yunnan province rather than model results. We agree that it is speculative and the model results cannot directly support this hypothesis. We have now clarified this in the paper and added precipitation difference as another possible driver of the difference in the measured aerosols (see Fig. 11a, b) as follows:

"The annual mean concentrations of carbonaceous matter collected in 2016 were lower than those in 2015 for the two sites (Fig 11a, b), which could be partly due to the strict mitigation measures that improved the local atmospheric environment in the Mt. Yulong region. For example, the amount of soot emissions (2.44 Tg C) in 2016 were reduced by 21.76% compared to 2015 in Yunnan province (http://www.zhb.gov.cn/), where the Mt. Yulong locates. Other factors likely contributed to the inter-annual difference as well. The amount of precipitation is important to determine aerosol wet removal from the atmosphere during the transport (e.g., Wang et al., 2015). The stronger precipitation in 2016 than in 2015 at Mt. Yulong and GHZ (Fig. 11a, b) partly explains the smaller carbonaceous aerosols mass concentrations in 2016."

4. The current form of the abstract is informative but frequently distract my focus. Please show the readers your most important and exciting findings.

Response: We have significantly reduced the length of the abstract to just highlight the most important results and findings. Below is the revised abstract: "Deposition and accumulation of light-absorbing carbonaceous matters on glacier surfaces can alter the energy balance of glaciers. In this study, two years (December 2014 to December 2016) of continuous observations of carbonaceous aerosols in glacierized region of Mt. Yulong and Ganhaizi (GHZ) basin are analyzed. The average black carbon (BC) and organic carbon (OC) concentrations were 1.51±0.93 and 2.57±1.32 g m-3, respectively. The average secondary OC (SOC) concentration, quantified using BC-tracer method, was 1.67±1.15 g m-3. Although the annual mean OC/BC ratio was 2.45±1.96, monthly

mean BC concentrations during the post-monsoon season were even higher than OC in the high altitudes (approximately 5000 m asl) of Mt. Yulong. Strong photochemical reactions and local tourism activities were likely the main factors inducing high OC/BC ratios in the Mt. Yulong region during the monsoon season. The mean mass absorption efficiency (MAE) of BC, measured for the first time in Mt. Yulong, at 632 nm with a thermal-optical carbon analyzer under the filter-based method, was 6.82$\pm$0.73 m2 g-1, comparable with the results from other studies. Strong seasonal and spatial variations of BC MAE were largely related to the OC and SOC abundance. Source attribution analysis using a global aerosol-climate model, equipped with a BC source tagging technique, suggests that East Asia emissions, including local sources, have the dominant contribution (over 50%) to annual mean near-surface BC in the Mt. Yulong area. There is also a strong seasonal variation in the regional source apportionment. South Asia has the largest contribution in BC emission during the pre-monsoon season, while East Asia dominates the monsoon season and post-monsoon season. Results in this study have great implications for accurately evaluating the influences of carbonaceous matter on glacial melting and water resource supply in glacierization areas."

Specific comments: 1. Title: overstated as pointed out above.

Response: We have changed the title of this manuscript to "Seasonal variation and light absorption property of carbonaceous aerosol in a typical glacier region of the southeastern Tibetan Plateau".

2. P3, line 8: missing reference.

Response: We have added at least one reference for each point as follows. "... visibility (Park et al., 2003), atmospheric radiative balance (Bond et al., 2013; Schuckmann et al., 2016), and the surface albedo of snow and ice (Gertler et al., 2016; Kaspari et al., 2014; Niu et al., 2017a, b)."

3. P4, line 1: specify the "absorbing aerosols".

Response: The "absorbing aerosols" denote aerosols containing BC, brown carbon, and/or dust that can absorb solar radiation. Below is the revised text. "... absorbing aerosols (e.g., BC, dust, and brown carbon)"

4. Line 7: partially correct.

Response: We have revised this sentence to the following: EC is usually denoted by BC (e. g., Cheng et al., 2011a; Ming et al., 2013; Xu et al., 2009b), which is assumed to have a nearly elemental composition. This is most suitable to diesel soot or lamp black, where BC has a chemical composition close to EC (Andreae and Gelencsér, 2006).

5. Line 13: overstated.

Response: We revised the previous sentence to the following: "This is closely related to water resources for a large population of local habitants in South Asia."

6. Line 17: For BC particles,

Response: We have added "," after "For BC particles" in the revised manuscript.

7. P6, line 27: specify the number of samples collected at the two sites.

Response: We added the number of samples collected at the two sites in the current submitted manuscript. "The number of aerosol samples collected at Mt. Yulong and GHZ was 117 and 120, respectively."

8. P7, line 3-5: this is nonsense if you have no attempt to give more details.

Response: We agree that this sentence doesn't add much useful information to the paper, so we have decided to remove this sentence.

9. Line 9: specify the temperature protocol you adopted.

We adopted the improved U.S. Interagency Monitoring of Protected Visual Environments (IMPROVE)-A thermal/optical reflectance protocol (Niu et al., 2017b). The residence time of each heating step was determined by the stabilization of the carbon signal. This has now been added to the text.

10. Line 20: the authors suggested different values of (OC/BC)min should be used to estimate SOC for different dataset (pre- or after-monsoon season), but I didn't see the discussion about data processing.

Response: We have now added a discussion on different values of (OC/BC)min in section 3.1 of the revised manuscript as follows. "Note that the minimum value of OC/BC ratio is used in the SOC calculation, as described in Eqs. (1) and (2). It varies greatly among different seasons (i.e., 0.38, 0.71, 0.42 and 0.35 for winter, pre-monsoon, monsoon, and post-monsoon in Mt. Yulong, respectively), so seasonal minimum values are used to estimate SOC concentrations in the corresponding seasons."

11. P8, line 1-4: given that "there is no standard or more perfect method of OCsec estimation (Yu et al., 2007)", how can you claim that "BC-tracer method employed here is reliable in determining SOC concentrations."

Response: We have now revised the statement to the following: "The semi-empirical (BC-tracer technique) estimates of OCsec after screening for outliers are comparable to the empirical estimates, and sometimes are even superior to empirical estimates. Thus, we believe the BC-tracer method employed here is acceptable in determining SOC concentrations."

12. P10, line 29: needs solid evidence.

Response: We have now provided the following evidence to support the statement in the revised manuscript. "An obviously higher BC and OC concentrations were found in the post-monsoon season at the Mt. Yulong site and in winter season at the GHZ site (Figs. 4 and 5) when wet removal by precipitation is inefficient. This suggests the importance of seasonal changes in sources (Carrico et al., 2003; Cong et al., 2015b; Wan et al., 2017). In addition, OC/BC ratio was usually employed to evaluate the

combustion fuel sources. Previous studies reported that the global mean of OC/BC by biomass burning was higher than fossil fuel burning (Bond et al., 2004; Cao et al., 2010; Liousse et al., 1996). Vehicle emissions from numerous touring buses in the GHZ basin play an important role in the variations of OC/BC ratios."

13. P11, line 9: NCO-P?

Response: NCO-P stands for Nepal Climate Observatory-Pyramid (NCO-P). We have noted this in the revised manuscript.

14. Line 13-15: needs solid evidence.

Response: This was based on conclusions of previous studies that are mentioned later in the text. We have decided to remove this statement here.

15. Line 17: keep a constant significance digit.

Response: Done as suggested for the entire manuscript.

16. Line 17-20: lacking of logic.

Response: We have revised the statement and provided solid evidence to support our argument in the revised manuscript. "It is quite likely that frequent rainfall events with occasional dust (e.g., Dong et al., 2011; Niu et al., 2014) from anthropogenic activities (Shrestha et al., 2000) during the monsoon in 2015 are responsible for this unusual phenomenon (i.e. relatively high content in monsoon season in 2015)."

17. Line 21: the annual average concentrations of OC and SOC.

Response: This sentence has been revised as follows. "... the annual average concentrations of OC and SOC, $3.50 \pm 1.50$ ïA■g m-3 and $2.20 \pm 1.40$ ïA■g m-3, respectively, ..."

18. Line 23-25: any solid evidence?

Response: We don't have solid evidence, but we revised the text to reflect our hypothesis/assumption on this: "Remote sources are likely to have a similar impact on aerosols over the two sites that are fairly close to each other. Therefore, the additional OC and SOC in the GHZ basin was more likely contributed by local sources such as fossil fuel (vehicle) emissions associated with the frequent and intense tourism activities, which is totally different from that of Mt. Yulong."

19. P12, line3-5: missing reference or supporting evidence.

Response: We have rewritten this sentence to actually support the previous hypothesis on the impact of local vehicle emissions. "... which is consistent with the assumption that the extensive vehicle emissions contributed to SOC formation in the GHZ basin. The monsoon season is also the peak tourism season, and the strong solar radiation in summer (Fig. 2) can enhance the yield of SOC through photochemical reactions (Antony et al., 2011; Schneidemesser et al., 2009; Wan et al., 2017; Niu et al., 2017b, c)."

20. Line 6: "somewhat" should be avoided as much as possible in a scientific paper.

Response: We deleted the word "somewhat" in the revised manuscript.

21. Line 5-8: needs solid evidence.

Response: We have now made it clear in the revised manuscript. "...The seasonal variation of carbonaceous aerosols found in the GHZ basin was different from that found in Mt. Yulong. It is likely related to the distinct elevation difference (nearly 1,500 m) and different amount of local human activities (here mainly referring to tourism related activities) between the Mt. Yulong and the GHZ sampling sites. The GHZ site location is close to a parking lot for private vehicles and touring buses and a visitor service center that involves food cooking. These tourism activities can contribute to local emissions of carbonaceous aerosols and precursor gases for SOC (Borrego et al., 2000; Cong et al., 2015a; Shi et al., 2017). However, we don't have direct observational evidence to support this."

22. Line 21-26: needs solid evidence.

Response: Please see our response to the previous comments on similar issues. There is no solid evidence for the statement, so we have removed it here.

23. P13, line 18: any reference?

Response: We have added two references to the statement. "In addition, brown carbon (BrC) appearing in the particle mixture can decrease the BC MAE (Jeong et al., 2004; Hecobian et al., 2010).

24. Line 25: "apparent spatial discrepancy"?, I don't think so.

Response: We have rephrased it to "a discernable small-scale spatial variation".

25. Line 28-30: unconvinced evidence. Any reference?

Response: We agree that the evidence is not convincing. The statement has been rewritten as "In addition to the elevation difference, other potential factors such as the more tourism activities near GHZ than at Mt. Yulong could partly account for the difference, but there is no observational evidence to support this."

26. P14, line 2-5: I suggest the authors to explore the effects of precipitation on the difference of carbonaceous aerosols mass loadings measured in different years.

Response: Following the great suggestion, we analyzed the annual total precipitation amount (mm) in 2015 and 2016 observed at the Mt. Yulong and GHZ sites and added it to Fig. 11a, b. We have also included a discussion on the possible impact of precipitation on the difference of carbonaceous aerosols mass loading between 2015 and 2016 at the Mt. Yulong and GHZ sites as follows: "Other factors likely contributed to the inter-annual difference as well. The amount of precipitation is important to determine aerosol wet removal from the atmosphere during the transport (e.g., Wang et al., 2015). The stronger precipitation in 2016 than in 2015 at Mt. Yulong (Fig. 11a) and GHZ (Fig. 11b) partly explains the smaller carbonaceous aerosols mass concentrations in 2016."

[Figure]

Fig. 11 (a) and (b)

27. Line 12-16: I am not sure if these results in previous studies were derived from TSP samples.

Response: We confirmed that the results in previous studies were derived from TSP samples.

28. P15, line 25-27: needs more solid evidences.

Response: This statement is based on the climate model simulation that used the 2010-2014 emissions. As discussed in our response to the major comment #3, we don't mean to attribute sources for BC measured at the sampling sites during 2015-2016, but rather to characterize the seasonal variation in the climatological mean source attribution. We have revised the sentence to clarify a little more. "According to our climate model results, emissions (2010-2014) from East Asia, including local sources accounted in the emission dataset, have a dominant contribution to the near-surface BC in the Mt. Yulong area during the monsoon (78%) and post-monsoon (68%) seasons, as well as the winter season (53%)."

[revised manuscript text omitted]

Fig. 11 (a) and (b)

**Fig. 1.** Fig. 11 (a) (b)

---

## Editor Comment (EC1) · X. Xu (Editor) · 8 Feb 2018

Dear authors,

I can only see you response to anonymous referee #2 in your AC1. Responses to all referees should appear in the interactive discussion. Please post your author response to anonymous referee #2.

———————————————

---

## Author Comment (AC2) · 8 Feb 2018

RC 2 Anonymous Referee #1

Comments to the Author This manuscript presented a topic with spatial and temporal variability and light absorption property of carbonaceous aerosols (OC and BC) in a typical glacierization region of Tibetan plateau. However, there are several key issues that authors should address and enhance/clarify (see below):

(a) The annual mean OC/BC ratio was found to be highest in the monsoon season and lowest in winter. The author needs to mention, which types of photochemical reactions

are involved that increased OC/BC ratio. Generally, photochemical reaction changing their physical and chemical properties from the original molecule of the substance. When this occurs, these molecules tend to form a new structure, or combine with each other or other molecules. This may change the state of OC or BC or OC/BC ratios.

Response: Following the great suggestion, we have added a discussion to the manuscript as follows. "The aging process of BC (or soot) resulting from photochemical oxidation by molecular O2 and the photooxidation of OC (Han et al., 2012) were likely involved and increased OC/BC ratio. Photochemical reaction can change their physical and chemical properties from the original molecule of the substance (alkyne C-H ($\equiv$C-H) and aromatic C-H (Ar-H)) (Kirchner et al., 2000; Cain et al., 2010). When this occurs, these molecules tend to form a new structure by combining with each other or with other molecules (carbonyl C=O and ether C-O) (Daly and Horn, 2009; Cain et al., 2010; Nieto-Gligorovski et al., 2008), which may change the state of OC, BC or OC/BC ratios. The photochemical oxidation by O2 under sunlight is an important aging process for BC (Han et al., 2012)."

(b) Author mentioned that tourism activities in monsoon season were attributed to relatively high OC/BC ratios in Mt. Yulong region, especially in GHZ basin. Firstly, author did not define tourism activities in whole manuscript. Second, how do author think "tourism activities" term used for vehicles or buses emission. Please explain in scientific literature that indicates tourism related activities contributed to OC, BC emissions with reference to justify the statement.

Response: Following the suggestion, we have now explained a little more about how the tourism related activities contributed to OC, BC emissions along with some references (i.e. Borrego et al., 2000; Cong et al., 2015a; Shi et al., 2017). Below is the revised text: "...The seasonal variation of carbonaceous aerosols found in the GHZ basin was different from that found in Mt. Yulong. It is likely related to the distinct elevation difference (nearly 1,500 m) and different amount of local human activities (here mainly referring to tourism related activities) between the Mt. Yulong and the GHZ

sampling sites. The GHZ site location is close to a parking lot for private vehicles and touring buses and a visitor service center that involves food cooking. These tourism activities can contribute to local emissions of carbonaceous aerosols and precursor gases for SOC (Borrego et al., 2000; Cong et al., 2015a; Shi et al., 2017). However, we don't have direct observational evidence to support this."

(c) In the result and discussion part "human activities" creating difference in Mt. Yulong and GHZ basin. This is a major observation indicated by author, but there is no concrete evidence available to support the results particularly what types of human activities?

Response: Human activities in our manuscript mainly refer to local tourism related activities, including the use vehicles and food cooking near the sampling site. This has been clarified in the text. Please also refer to our response to the comment (b) for more details added to the revised manuscript.

(d) Comments: Authors stated that "A large amount of biomass burning emissions in the high atmosphere (around 5000 m asl) in Mt. Yulong were probably long range transported from source regions". This observation is supported by the higher BC concentration than OC especially during monsoon season to post monsoon season. How do authors assume this observation without performing Trajectory analysis with HYSPLIT model? In order to find the transport of pollutant emission from biomass burning, author needs to perform trajectory analysis during the study period to support the results for Dec. 014 and Dec. 2016.

Response: Following the suggestion, we performed trajectory analysis with the HYS-PLIT model. From the monsoon season to post monsoon season, the trajectory of air mass reaching the sampling location changed much. In the monsoon season, the air mass (and pollutants) mainly originated from southwest and southeast monsoons, while in the post monsoon season it mainly came from the west. However, we still cannot verify the source type. In response to similar comments from RC1, we have

revised the text and removed the unverified statements.

Fig. 7 Seven-day backward trajectory analysis with HYSPLIT model (a) in the monsoon and (b) post monsoon seasons during the study period (Source ★ at 27.01 N, 100.20 E). The trajectories of air mass in the plot were the average of a few episodes. The two heights are the elevation of Mt. Yulong and GHZ, respectively.

(e) Authors selected experiment dates (December 2014-December 2016) for time series of meteorological parameters at Mt. Yulong. On the other hand, in model experimental part for CAM5, selected dates are 2010 to 2014, although the author stated the reason but for actual results synchronization, it should be similar dates to give clearer picture and what is base to select there years?

Response: we totally understand the concern here, which is similar to the major comment 3) of RC1. Please refer to our response there regarding the limitation and purpose of the climate model results. This has been clarified more in the revised manuscript.

(f) In result and discussion parts, author mentioned "seasonal changes in BC and OC sources (e.g., biomass burning vs. fossil fuel combustion) might play an important role for the variations of OC/BC ratios". How does seasonal change affect in OC/BC ratio? Explain the reason and factors involved in this change.

Response: We totally understand the concern here, which is similar to the comment 12) of RC1. Below is the revised text: "An obviously higher BC and OC concentrations were found in the post-monsoon season at the Mt. Yulong site and in winter season at the GHZ site (Fig. 4 and 5) when wet removal by precipitation is inefficient. This suggests the importance of seasonal changes in sources (Carrico et al., 2003; Cong et al., 2015b; Wan et al., 2017). In addition, OC/BC ratio was usually employed to evaluate the combustion fuel sources. Previous studies reported that the global mean of OC/BC by biomass burning was higher than fossil fuel burning (Bond et al., 2004; Cao et al., 2010; Liousse et al., 1996). Vehicle emissions from numerous touring buses in the GHZ basin play an important role in the variations of OC/BC ratios.

(g) How did author's calibrate the particulate sampling apparatus (TH150-Tianhong INST group)? And the quality assurance of this apparatus is not explained in manuscript.

Response: We calibrated the particulate sampling apparatus by weighting the mass of atmospheric gas filtrated by a vacuum pump, and recording the temperature of gas to calculate the virtual volume of gas during a specific collecting timescale. Then we made comparisons between the calculated value and the displayed volume of gas in the apparatus. In addition, we adjusted the flow of TSP (i.e. total suspended particulate), and recorded the meteorological parameters during the sampling period to timely calibrate the apparatus in case of inaccuracy. The quality assurance of this apparatus is explained in the revised manuscript: "The quality assurance of this apparatus (TH150-A) is demonstrated by the difference between manually calculated volume of gas and automatically recorded value. The volume of atmospheric gas was usually recorded automatically by the apparatus. The air (and suspended particulates) was sampled at a flow rate of 100 L min-1 with an accuracy of $\pm$2.5%, and each sample was collected for 24 h using a stable vacuum pump, which had a good quality and performance (e.g., running steadily with low noise level)."

Some minor revision is as follows: (a) In the abstract part line 2-3 write down the name for adjoining areas.

Response: In addressing a comment of RC1, we deleted this sentence (... adjoining areas) in the abstract focus just on the most important and exciting findings.

(b) In the introduction part line 1-2 add Bond et al. 2007 reference to support the statement.

Response: We have now added Bond et al. (2007) reference to support the statement.

(c) In the introduction part line 3‐4 use words as " human health and living species" instead of using "living organisms"

Response: We have changed the "living organisms" to "human health and living species" in the revised manuscript.

(d) In the introduction part line 4 reference is missing.

Response: We have now added references for these points. "... visibility (Park et al., 2003), atmospheric radiative balance (Bond et al., 2013; Schuckmann et al., 2016), and the surface albedo of snow and ice (Gertler et al., 2016; Kaspari et al., 2014; Niu et al., 2017a, b)."

(e) Page 4; line 4 Tibetan plateau (TP), abbreviation is already used in abstract part, so need to use full word.

Response: The sentence contains "Tibetan plateau (TP)" in the abstract was deleted, so the "TP" is now spelled out in the introduction, but we made sure that only the abbreviation is used thereafter.

(f) Page 5; line 12 please elaborate the model names that are not considered absorption by OC with reference.

Response: There are many of them, but we have now added one model name (CESM) and references as an example. CESM is a very popular community model that has many users, which is also used in this study. "However, light absorption by OC has not yet been taken into consideration in many climate models, e.g., various versions of the Community Earth System Model (CESM) (Flanner and Zender, 2006; Wang et al., 2013; Qian et al., 2015; Liu et al., 2016), ..."

(g) In material and method part page 5; Line 29. This statement needs reference to support the arguments.

Response: This sentence doesn't add much useful information to the paper, so we have decided to remove this sentence.

(h) Page 6, line 13; use some other words instead of using "But".

Response: Changed to "However, it becomes . . .".

(i) Page 6; line 16, Is there any other pollution sources? if yes please mention and explain their sources.

Response: Yes, there are other minor pollution sources. We have now mentioned them in the revised manuscript. The following is the revised statement. "Besides the major emissions from tourist vehicles, there are some other limited pollution sources, such as agricultural waste burning, (open fire) biomass burning, and crustal aerosols (Niu et al., 2014, 2016) near the study sites."

(j) Page 9; line 20, write down "Then they" instead of using "They then"

Response: Changed as suggested.

(k) Page 10; line 20 reference is missing

Response: We have added some references to support the statement: It is quite likely that frequent rainfall events with occasional dust (e.g., Dong et al., 2011; Niu et al., 2014) from anthropogenic activities (Shrestha et al., 2000) during the monsoon in 2015 are responsible for this unusual phenomenon (i.e. relatively high content in monsoon season in 2015).

(l) Page 12; line 7 what type of human activities involved between Mt. Yulong and the GHZ sampling sites?

Response: The involved human activities are vehicle emissions and visitor services near the GHZ site but only sightseeing on Mt. Yulong. We have now made it clear in the revised manuscript. The following is the revised text. ". . .The GHZ site location is close to a parking lot for private vehicles and touring buses and a visitor service center that involves food cooking. These tourism activities can contribute to local emissions of carbonaceous aerosols and precursor gases for SOC (Borrego et al., 2000; Cong et al., 2015a; Shi et al., 2017). However, we don't have direct observational evidence to support this."

(m) Page 14; line 17 write down the full name for "QSMS".

Response: There was a typo in the abbreviation, which is supposed to be "QOMS". It stands for Qomolangma (Mt. Everest) Station for Atmospheric and Environmental Observation (Cong et al., 2015b).

(n) Page 17;line 11 add "in BC emission" after "contribution".

Response: Revised as suggested. "East Asia has a dominant contribution in BC emission in the monsoon and post-monsoon seasons, ..."

(o) Page 41; Fig. 11. Is there any major difference or key observation that author found in current study which is actually different from others. Please explain in the footnotes of Fig.11.

Response: We have added a key message from the comparison in the caption of Fig. 11 (Fig. 12 of the revised manuscript): "Annual mean BC concentrations at Mt. Yulong were rather low ($\sim$1.5 g m-3), while most of the compared BC concentrations (counted by number of measurements) were within the range of 1.0-2.5 g m-3.

[revised manuscript text omitted]

---

## Author Response (AR3)

**Response to referees' comments**

Journal: ACP

Title: Seasonal variation and light absorption property of carbonaceous aerosol in a typical glacier region of the southeastern Tibetan Plateau

5   Author(s): Hewen Niu et al.

MS No.: acp-2017-865

MS Type: Research article

Iteration: Major Revision

Special Issue: Study of ozone, aerosols and radiation over the Tibetan Plateau
10  (SOAR-TP) (ACP/AMT inter-journal SI)

We thank the referee and editor for their careful review, insightful comments and suggestions, which greatly helped improve our manuscript. Below are our point-by-point responses and detailed corrections/revisions made to the manuscript
15  accordingly.

Report #2

Suggestions for revision

20  Usually, the term "EC" should be used here instead of BC as thermal-optical method was applied here. I strongly suggest use EC in this paper. And I noticed that TSP samples were collected, I doubt that carbonate carbon (CC) may be important in coarse fractions. CC will influence on the results of OC and EC as well as the estimated SOC. SOC estimation has a large uncertainty when biomass burning
25  emissions were important. I suggest removing SOC part as such an estimation may be not convincing.

Response: We agree with the referee that the refractory constituent of the carbonaceous aerosols determined using the thermal-optical method should be called
30  "elemental carbon" rather than "black carbon", which describes the light-absorbing

part determined using optical methods or in aerosol-climate models. Following the suggestions, we have replaced "BC" with "EC" when describing our measurements and analysis in the revised paper (with "BC" kept for the general discussion and the modeling part).

According to the suggestion, we have also removed the SOC-related materials (analyses, results and figures) from the revised manuscript.

Comments from editor Xiaobin Xu

Comments to the Author:

Dear authors:

Your revised manuscript has been reviewed by two referees. Additional revisions are suggested by referee #2. Please improve your manuscript further by considering the suggestions of referee #2 and also following comments from me:

1. Page 5 line: Ssandradewi et al., 2008 is not given in the references list. A thorough check of citations is needed.

Response: We have added (Sandradewi et al., 2008) to the reference list, and thoroughly checked the citations in the manuscript.

2. Page 6 line 9 and page 13 line 8: change "monsoons season" to "monsoon season".

Response: Corrected.

3. Page 6 lines 11-15: I do not think you should give elevations and coordinates of that accurate, in particular on the glaciers.

Response: Following the suggestion, we have reduced the number of significant digits for the elevations and coordinates.

4. Page 6 line 30: change "atmospheric gas" to "air".

Response: Done.

5. Page 7 lines 2-3: "using a stable vacuum pump, which had a good quality and performance (e.g., running steadily with low noise level)." It is better to give pump's model and manufacturer information instead of these vague expressions.

Response: We have added the pump's model and manufacturer information in the revised manuscript as the following:
"..., and each sample was collected for 24 h using a Thomas pump (2628TE32, America)."

6. Page 11 lines 7-10: What do you mean by "photochemical ability of OC" and "little material available for external-coating of BC"?

Response: The phrase "photochemical ability of OC" in the previous manuscript was a mistake, which has been corrected in the revised manuscript. We meant to say that relatively weak solar radiation in winter would result in less production of organic compounds through photochemical reactions.
We have revised the original sentence to "Lower ratio of OC/EC in the other seasons (winter, pre-monsoon, post-monsoon) was probably due to the less photochemical production of secondary organic compounds as coating material on EC particles (Knox et al., 2009; Cappa et al., 2012)" in the revised manuscript.

7. Page 11 and lines 19-21: How can this happen?

Response: The monsoon season is the peak tourism season and winter season is the off-season in GHZ. Thus the amount of vehicle emissions from touring buses and private vehicles had seasonal difference, which we believe played an important role

in the seasonal variations of OC/EC ratios at the sampling site. Since we don't have direct observational evidence to support this, we have revised the statement to: "Seasonal differences in vehicle emissions from touring buses and private vehicles in the GHZ basin might have played an important role in the seasonal variations of OC/EC ratios".

8. Fig. 2 and some other figures: there are obviously some data gaps. You should not fill the gaps with lines.

Response: The data gaps are now unfilled with lines in the revised manuscript (i.e. Fig. 2, Fig. 4, and Fig. 5).

9. Figs. 9 and 10: displaying MAE data repeatedly in all plots is not necessary. If you are discussing the correlations of MAE with other quantities, it is better to show the data as scatter plots with regression results.

Response: This is an excellent suggestion. We have now shown the data as scatter plots with regression results in the revised manuscript. The following are included as new figures (Figs. 9 and 10).

[Figure]

Fig. 9

[Figure]

 Fig. 10

**References**

[revised manuscript text omitted]